# Conservation of the cooling agent binding pocket within the TRPM subfamily

Kate Huffer, Matthew CS Denley, Elisabeth V Oskoui[†], Kenton J Swartz*

Molecular Physiology and Biophysics Section, Porter Neuroscience Research Center, National Institute of Neurological Disorders and Stroke, National Institutes of Health, Bethesda, United States

## eLife Assessment

In this **valuable** study, Huffer et al posit that non-cold sensing members of the TRPM subfamily of ion channels (e.g., TRPM2, TRPM4, TRPM5) contain a binding pocket for icilin that overlaps with the one found in the cold-activated TRPM8 channel. After examining a body of TRP channel cryo-EM structures to identify the conserved site, this study presents **convincing** electrophysiological evidence supporting the presence of an icilin binding pocket within TRPM4. This study shows that icilin has modulatory effects on the TRPM4 channel and will be of direct interest to those working in the TRP-channel field, but it also has implications for studies of somatosensation, taste, as well as pharmacological targeting of the TRPM subfamily.

*For correspondence:
swartzk@ninds.nih.gov

Present address: [†]Imperial College London, London, United Kingdom

## Abstract

Transient receptor potential (TRP) channels are a large and diverse family of tetrameric cation-selective channels that are activated by many different types of stimuli, including noxious heat or cold, organic ligands such as vanilloids or cooling agents, or intracellular $Ca^{2+}$. Structures available for all subtypes of TRP channels reveal that the transmembrane domains are closely related despite their unique sensitivity to activating stimuli. Here, we use computational and electrophysiological approaches to explore the conservation of the cooling agent binding pocket identified within the S1–S4 domain of the Melastatin subfamily member TRPM8, the mammalian sensor of noxious cold, with other TRPM channel subtypes. We find that a subset of TRPM channels, including TRPM2, TRPM4, and TRPM5, contain pockets very similar to the cooling agent binding pocket in TRPM8. We then show how the cooling agent icilin modulates activation of mouse TRPM4 to intracellular $Ca^{2+}$, enhancing the sensitivity of the channel to $Ca^{2+}$ and diminishing outward-rectification to promote opening at negative voltages. Mutations known to promote or diminish activation of TRPM8 by cooling agents similarly alter activation of TRPM4 by icilin, suggesting that icilin binds to the cooling agent binding pocket to promote opening of the channel. These findings demonstrate that TRPM4 and TRPM8 channels share related ligand binding pockets that are allosterically coupled to opening of the pore.

## Introduction

The founding member of the transient receptor potential (TRP) channel family was cloned two decades after a *Drosophila trp* mutant was discovered to exhibit a transient receptor potential in electroretinogram recordings (*Cosens and Manning, 1969*; *Montell and Rubin, 1989*; *Wong et al., 1989*). Many other TRP channels have been subsequently identified in eukaryotes and classified into 10 subfamilies based on sequence similarity (*Himmel and Cox, 2020*; *Cabezas-Bratesco et al., 2022*). All TRP

**Figure 1.** Structures of vanilloid bound to TRPV1 and cooling agent bound to TRPM8. Side views of (**A**) TRPM8 bound to icilin (CPK) and Ca²⁺ (green sphere) and (**B**) TRPV1 bound to RTx (CPK). Intracellular views of (**C**) TRPM8 bound to icilin and Ca²⁺ and (**D**) TRPV1 bound to RTx.

channels are tetrameric cation channels that contain six transmembrane (TM) segments (S1–S6) per subunit (*Figure 1*), similar to voltage-activated K⁺ channels (*Cao, 2020*; *Huffer et al., 2020*). However, individual TRP channel family members respond to multiple different types of stimuli, including chemical ligands, temperature, voltage, membrane lipids, and mechanical deformation, and they often function as coincidence detectors that integrate multiple stimuli (*Chuang et al., 2004*). TRP channels can be regulated by G-protein pathways, and in many cases may function as receptor-operated channels, including the founding member in *Drosophila* photoreceptors (*Plant and Schaefer, 2003*; *Vazquez et al., 2004*; *Kukkonen, 2011*; *Shalygin et al., 2021*; *Rohacs, 2024*). Two of the most extensively studied TRP channels are TRPV1 and TRPM8, which in mammals function as detectors of noxious heat (*Caterina et al., 1997*; *Tominaga et al., 1998*; *Caterina et al., 2000*) and noxious cold

(*McKemy et al., 2002*; *Peier et al., 2002*; *Bautista et al., 2007*; *Dhaka et al., 2007*), respectively. Although the mechanisms of temperature sensing in TRPV1 and TRPM8 remain incompletely understood (*Bandell et al., 2006*; *Grandl et al., 2008*; *Grandl et al., 2010*; *Clapham and Miller, 2011*; *Qin, 2014*; *Jara-Oseguera et al., 2016*; *Liu and Qin, 2017*; *Zhang et al., 2018a*; *García-Ávila and Islas, 2019*; *Diaz-Franulic et al., 2021*; *Lu et al., 2022*; *Yeh et al., 2023*), there is consensus about where vanilloids like capsaicin bind to activate TRPV1 and cooling agents such as icilin bind to activate TRPM8 (*Cao et al., 2013b*; *Liao et al., 2013*; *Gao et al., 2016*; *Yin et al., 2018*; *Yin et al., 2019a*; *Yin and Lee, 2020*; *Zhang et al., 2021*; *Zhao et al., 2022*).

Vanilloids and cooling agents were originally proposed to bind to similar regions of TRPV1 and TRPM8 because mutations affecting channel activation by these compounds could be found in similar TM regions (*Chuang et al., 2004*). However, structures solved using cryogenic electron microscopy (cryo-EM) have since revealed that vanilloids bind to a membrane-facing pocket positioned at the interface between the pore domain formed by S5–S6 and the peripheral S1–S4 domains (*Cao et al., 2013b*; *Liao et al., 2013*; *Gao et al., 2016*; *Zhang et al., 2021*), whereas cooling agents bind within a pocket formed entirely by the S1–S4 helices that opens to the cytoplasm (*Yin et al., 2018*; *Yin et al., 2019a*; *Yin and Lee, 2020*; *Figure 1*). The vanilloid binding pocket in TRPV1 can be occupied by vanilloid agonists and antagonists (*Cao et al., 2013b*; *Liao et al., 2013*; *Gao et al., 2016*; *Kwon et al., 2021*; *Zhang et al., 2021*; *Kwon et al., 2022*; *Neuberger et al., 2023*), in addition to membrane lipids that have been proposed to stabilize a closed state of the channel (*Cao et al., 2013a*; *Gao et al., 2016*; *Zhang et al., 2021*; *Arnold et al., 2024*). Cryo-EM maps for all five available TRPM8 structures with the cooling agent icilin bound (6nr3, 7wrc, 7wrd, 7wre, and 7wrf) show a large non-protein density consistent with icilin nestled within the S1–S4 domain (*Yin et al., 2018*; *Yin et al., 2019a*; *Yin and Lee, 2020*; *Zhao et al., 2022*; *Figure 1*). This cooling agent binding pocket is lined by residues where mutations are known to modulate sensitivity to icilin and/or the cooling agent menthol (*Chuang et al., 2004*; *Bandell et al., 2006*; *Yin et al., 2019a*; *Yin and Lee, 2020*; *Zhao et al., 2022*), supporting the assignment of this density to icilin during model building (*Figure 2*). Other TRPM8 ligands have also been observed in this pocket, including cooling agents WS-12 (6nr2) and C3 (8e4l, 8e4m, 9b6h) and the antagonists AMTB (6o6r, 9b6g), TC-I 2014 (6o72, 9b6e, 9b6h, 9b6i), and AMG2850 (9b6f) (*Diver et al., 2019*; *Yin et al., 2019a*; *Yin and Lee, 2020*; *Yin et al., 2022*; *Zhao et al., 2022*; *Yin et al., 2024*). Interestingly, other TRP channel structures from the Canonical and Vanilloid subfamilies also show small molecule ligands bound to a similar location within the S1–S4 domain (7b0s, 7b05, 7wdb, 7d4p, 6uza, 7dxg, 6dvy, 6dvz, 7ras, 7rau, 6pbe, 6d7o, 6d7q, 6d7t, 6d7v, 6d7x) (*Singh et al., 2018a*, *Singh et al., 2018b*; *Hughes et al., 2019*; *Bai et al., 2020*; *Vinayagam et al., 2020*; *Neuberger et al., 2021*; *Song et al., 2021*; *Guo et al., 2022*), raising the possibility that this pocket may be a hot spot for ligand modulation across the TRP channel family.

The cytoplasmic entrance to the cooling agent binding pocket in TRPM8 also contains a binding site for a $Ca^{2+}$ ion (6nr3, 7wrb, 7wrc, 7wrd, 7wre, 7wrf, 8e4l, 8e4m, 6o77, 9b6i, 9b6j, 9b6k) (*Figure 1*; *Diver et al., 2019*; *Yin et al., 2019a*; *Yin et al., 2022*; *Zhao et al., 2022*; *Yin et al., 2024*). While $Ca^{2+}$ alone has not been described to modulate TRPM8 activity, it is an essential cofactor for activation of TRPM8 by icilin (*Chuang et al., 2004*), and several residues, including E782 in S2 and D802 in S3 of TRPM8 from *Ficedula albicollis* (faTRPM8) and mouse (mTRPM8), are located within 4 Å of both the $Ca^{2+}$ ion and the icilin molecule (*Yin et al., 2019a*; *Zhao et al., 2022*). Mutations in N799 and D802 in S3 of TRPM8 from rats (rTRPM8) and humans (hTRPM8) have been implicated in selective loss of icilin sensitivity (*Chuang et al., 2004*; *Winking et al., 2012*; *Kühn et al., 2013*; *Beccari et al., 2017*). Because N799 does not contact the icilin molecule directly, it presumably perturbs icilin sensitivity by disrupting binding of the obligate cofactor, $Ca^{2+}$. Interestingly, the equivalent $Ca^{2+}$-binding site has been observed in several other TRPM channel structures (6bqv, 6co7, 6d73, 6drj, 6nr3, 6o77, 6pkx, 6pus, 6puu, 7mbq, 7mbs, 7mbu, 7mbv, 7wrb, 7wrc, 7wrd, 7wre, 7wrf) (*Autzen et al., 2018*; *Huang et al., 2018*; *Zhang et al., 2018b*, *Diver et al., 2019*; *Huang et al., 2019*; *Yin et al., 2019a*; *Yin et al., 2019b*; *Ruan et al., 2021*; *Zhao et al., 2022*), including TRPM2, TRPM4, and TRPM5, for which intracellular $Ca^{2+}$ is the key physiological agonist, and where mutations to the $Ca^{2+}$-coordinating residues have been shown to alter channel activation by intracellular $Ca^{2+}$ (*Guo et al., 2017*; *Autzen et al., 2018*; *Zhang et al., 2018b*; *Yamaguchi et al., 2019*). Structures from other TRP channel subfamilies—including Canonical (5z96, 6aei, 6jzo, 7b05, 7b0j, 7b0s, 7b16, 7b1g, 7d4p, 7d4q, 7dxb, 7dxf, 7e4t, 7wdb) (*Duan et al., 2018*; *Vinayagam et al., 2018*; *Duan et al., 2019*; *Song et al., 2021*; *Guo et al.,*

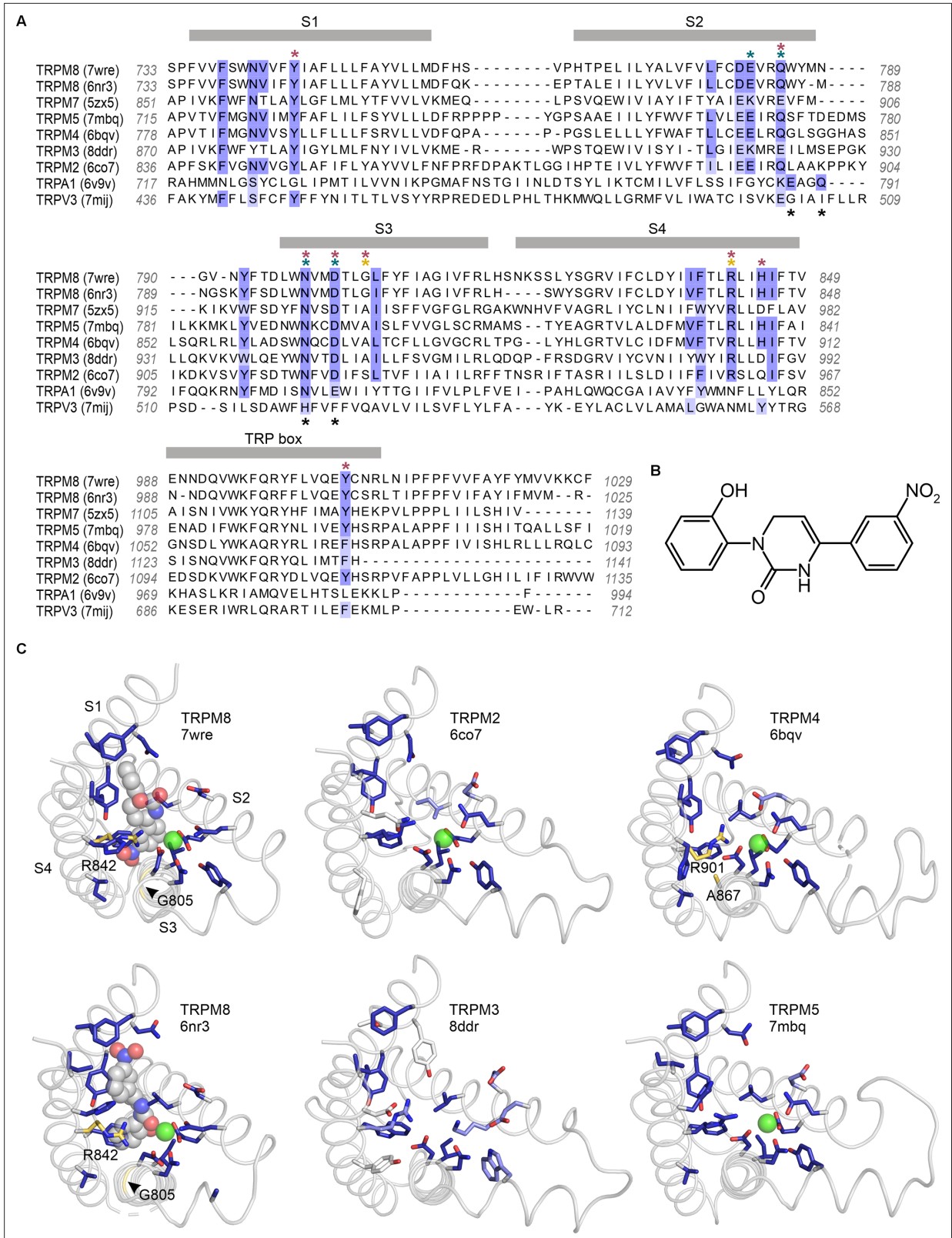

**Figure 2.** Sequence and structure conservation of the icilin binding pocket in TRPM and TRPA channels. (**A**) Structure-based sequence alignment of S1–S4 peripheral domains and transient receptor potential (TRP) helix of selected TRP channel structures, with residues contributing to the icilin binding pocket in TRPM8 structures (7wre and 6nr3) highlighted in blue. The equivalent residues in other channels are colored according to the alignment quality score calculated from multiple sequence alignments, where highly conserved residues are color blue and poorly conserved residues are colored

*Figure 2 continued on next page*

*Figure 2 continued*

in white. Alignment quality score calculated in Jalview based on BLOSUM 62 scores (*Henikoff and Henikoff, 1992*). Teal asterisks indicate $Ca^{2+}$-coordinating residues in structures of TRPM channels. Black asterisks indicated $Ca^{2+}$-coordinating residues in TRPA1. Red asterisks indicated residues where mutation influence cooling agent sensitivity in TRPM8. Gold asterisks indicate residues mutated in the present study. (**B**) Chemical structure of icilin. (**C**) S1–S4 residues contributing to the icilin binding pocket in TRPM8 structures (7wre and 6nr3) are shown as blue licorice, viewed from the intracellular side of the membrane as in *Figure 1C*, with the TRP box omitted for clarity. Cooling agent binding pocket mutations used in the present study are shown with carbon atoms colored gold and labeled in TRPM8 and TRPM4, and the equivalent residues in other channels are colored based on the alignment quality score, as in panel A. 7wre is mTRPM8, 6nr3 is faTRPM8 containing the A805G mutation, 6co7 is *Nematostella vectensis* TRPM2, 8ddr is mTRPM3, 6bqv is hTRPM4, and 7mbq is zebra fish TRPM5. Sequence identity between residues within the icilin binding pocket of TRPM8 and corresponding residues in the other TRP channel is as follows: TRPM5 (94%), TRPM4 (89%), TRPM2 (78%), TRPM3 and TRPM7 (44%), TRPA1 (22%), and TRPV3 (11%).

The online version of this article includes the following figure supplement(s) for figure 2:

**Figure supplement 1.** S1–S4 residues contributing to the icilin binding pocket in TRPM8 structures 7wre and 6nr3 are shown as blue licorice, viewed from the intracellular side of the membrane, with the transient receptor potential (TRP) box omitted for clarity.

*2022*; *Yang et al., 2022*), Ankyrin (6v9v, 6v9w, 7or0, 7or1) (*Zhao et al., 2020*), and Polycystin (7d7e, 7d7f) (*Su et al., 2021*) families—also contain cryo-EM density attributed to ions in this location. As in the cooling agent binding pocket, the presence of this ion-binding pocket within the S1–S4 domains across the TRP channel family indicates that it is another key regulatory location.

The structural observations of related binding pockets for different activators across the TRP channel family may be connected to the conservation of mechanisms of channel activation between channels that exhibit different sensitivity to activating stimuli. For example, TRPV2 and TRPV3 are insensitive to the TRPV1-specific high-affinity vanilloid agonist Resiniferatoxin (RTx), but sensitivity to this vanilloid can be readily engineered into both TRPV channels by mutating only a few key residues within the vanilloid binding site (*Yang et al., 2016*; *Zhang et al., 2016*; *Zubcevic et al., 2018*; *Zhang et al., 2019*), revealing that the mechanisms responsible for coupling occupancy of the vanilloid site to channel opening are conserved in these three TRPV channels. In the present study, our aim was to explore the extent to which the cooling agent binding pocket described in TRPM8 is conserved in other TRPM channels and to then determine whether cooling agents can modulate channel opening in channels other than TRPM8. Our results suggest that the cooling agent binding pocket is well-conserved in several TRPM channels that were not previously reported to be sensitive to cooling agents and we find that icilin can bind to this well-conserved site in TRPM4 and promote opening of the pore by intracellular $Ca^{2+}$ and alter the outwardly rectifying properties of the channel.

## Results

### Identification of residues lining the icilin binding pocket

We first investigated the icilin binding pocket using available structures of TRPM8 in complex with icilin and its obligate cofactor, $Ca^{2+}$ (*Yin et al., 2018*; *Yin et al., 2019a*; *Yin and Lee, 2020*; *Zhao et al., 2022*). We defined the icilin binding pocket to include residues located near the icilin molecules in the available icilin-bound TRPM8 structures. With sufficient structural resolution, icilin's asymmetric arrangement of a hydroxyl group on one terminal benzene ring and a nitro group on the other benzene ring (*Figure 2*) should facilitate identification of the ligand's physiological binding pose. Unfortunately, the electron densities attributed to icilin lack sufficient definition to fit these asymmetric functional groups in all but one of the five icilin-bound structures, even though the S1–S4 helices have relatively high local resolution in these structures. Interestingly, the structure with the most asymmetric icilin density (7wrd, with 2.98 Å overall nominal resolution) is not the highest resolution structure (7wre, with 2.52 Å overall nominal resolution). Previous comparison of these two binding poses has indicated that both poses are similarly plausible (*Palchevskyi et al., 2023*), although the energy minimization performed in this study was conducted in the absence of the obligate cofactor, $Ca^{2+}$, which may be important for stabilizing icilin binding. We therefore decided to consider both icilin binding poses as possibly valid in defining the icilin binding pocket.

We identified the residues lining the icilin binding pocket based on residue proximity to icilin and on the ligand–protein interaction fingerprint generated by PoseFilter (*Williams and Kalyaana-moorthy, 2021*), which classifies interactions between the protein and ligand based on their chemistry

in addition to proximity. We chose to use the most inclusive definition of binding pocket lining residues, counting any residue that was within 4 Å of the icilin molecule in either pose (6nr3 or 7wre), which also included all interacting residues identified by PoseFilter (*Figure 2*). In addition, we included the $Ca^{2+}$-coordinating residues in the binding pocket because $Ca^{2+}$ is an essential cofactor and mutations in $Ca^{2+}$-coordinating residues have been described to specifically disrupt icilin sensitivity in TRPM8 (*Chuang et al., 2004*; *Winking et al., 2012*; *Kühn et al., 2013*; *Zhao et al., 2022*).

## Conservation of the icilin binding site in the Melastatin subfamily

To compare identified icilin binding pocket residues to other TRP channels, we utilized a structural alignment approach as described previously (*Huffer et al., 2020*), but expanding the scope of analysis to include a total of 264 structures of TRP channels determined to date. Comparing other Melastatin subfamily channels to TRPM8 revealed unexpectedly high conservation in the residues lining the cooling agent binding pocket in TRPM2 (78% identical), TRPM4 (89% identical) and TRPM5 (94% identical) (*Figure 2*), channels that have not previously been reported to be sensitive to icilin. Mutations of residues in the structurally identified cooling agent binding pocket located within the S1–S4 domain of rTRPM8, including those corresponding to Y745 in S1, Q785 in S2, N799 and D802 in S3, R842 and H845 in S4, and Y1005 in the TRP box, are known to functionally influence cooling agent sensitivity in mTRPM8, rTRPM8, hTRPM8, *Parus major* TRPM8 (pmTRPM8) or faTRPM8 (*Chuang et al., 2004*; *Bandell et al., 2006*; *Voets et al., 2007*; *Malkia et al., 2009*; *Winking et al., 2012*; *Kühn et al., 2013*; *Beccari et al., 2017*; *Yin et al., 2018*; *Diver et al., 2019*; *Yin et al., 2019a*; *Yin and Lee, 2020*; *Plaza-Cayón et al., 2022*; *Zhao et al., 2022*). These important residues (*Figure 2A*; red asterisks), along with other residues located within 4 Å of icilin or $Ca^{2+}$ in TRPM8 (*Figure 2A*; blue highlighting), are highly conserved between TRPM8 and TRPM2, TRPM4, and TRPM5. These channels also show structural similarity in the shape of the pocket and orientation of the equivalent residues (*Figure 2C*). Although a recently reported structure of TRPM4 prepared at physiological temperatures reveals interesting structural changes within the intracellular melastatin domains compared to earlier structures, the structure of the cooling agent binding pocket is very similar (*Hu et al., 2024*). The one notable and important difference near the cooling agent binding pocket between TRPM8, TRPM2, TRPM4, and TRPM5 is at the position corresponding to G805 in rTRPM8, a position that is conserved in mammalian TRPM8 channels that are sensitive to icilin, but is substituted by an Ala in avian TRPM8 channels that are insensitive to icilin (*Chuang et al., 2004*). Because this critical position in TRPM8 is an Ala in most other TRPM channels (*Figure 2*), we predicted that TRPM2, TRPM4, and TRPM5 might be insensitive to icilin but could be rendered sensitive to icilin by substituting a Gly, similar to the icilin-sensitizing mutants engineered into avian TRPM8 channels (*Chuang et al., 2004*; *Yin et al., 2019a*). In the case of TRPM3 and TRPM7, in addition to lacking the critical Gly residue, there is less conservation in the icilin- and $Ca^{2+}$-adjacent residues (44% identity in both cases) (*Figure 2*), suggesting that these channels may not be sensitive to icilin. TRPM1 and TRPM6, which were not included in the structural alignment because structures are not available, were predicted to behave similar to TRPM3 and TRPM7 based on conventional sequence alignments in the S1–S4 region (not shown). Interestingly, TRPA1 and TRPV3, which have both been previously described to be sensitive to icilin (*Story et al., 2003*; *Doerner et al., 2007*; *Sherkheli et al., 2010*; *Sherkheli et al., 2012*; *Billen et al., 2015*), do not show homology in the equivalent binding pocket (*Figure 2*; *Figure 2—figure supplement 1*; see Discussion).

## Characterization of TRPM4 sensitivity to icilin

Based on the observed structural conservation, we considered attempting to engineer icilin sensitivity into TRPM2, TRPM4, or TRPM5. TRPM2 requires co-activation by intracellular $Ca^{2+}$ and ADP ribose (ADPR) and its permeability to $Ca^{2+}$ alters the local intracellular $Ca^{2+}$ concentration (*Perraud et al., 2001*; *Sano et al., 2001*), making TRPM2 more challenging to study. In contrast, TRPM4 and TRPM5 are monovalent-selective and are activated by intracellular $Ca^{2+}$ binding to the conserved $Ca^{2+}$-binding site near the cooling agent binding pocket (*Launay et al., 2002*; *McKemy et al., 2002*; *Hofmann et al., 2003*; *Liu and Liman, 2003*; *Prawitt et al., 2003*; *Story et al., 2003*; *Andersson et al., 2004*; *Chuang et al., 2004*; *Yamaguchi et al., 2019*). In both TRPM4 and TRPM5, $Ca^{2+}$ dependence is also conferred by a second, intracellular $Ca^{2+}$-binding site unrelated to the $Ca^{2+}$-binding site near the cytoplasmic entrance of the cooling agent binding pocket in TRPM8 (*Ruan et al., 2021*; *Hu*

*et al., 2024*; *Karuppan et al., 2024*). At the time this study was initiated, the presence of this second $Ca^{2+}$ site in TRPM4 was not appreciated, so we chose to focus on TRPM4 as a simpler system, based on the fact that it is relatively impermeable to $Ca^{2+}$ (*Launay et al., 2002*) and the assumption that activation is controlled by a single stimulus binding to a single binding site in each subunit. TRPM4 is also widely expressed in the body and plays important physiological roles in the cardiovascular, immune, endocrine, and nervous systems (*Hasan and Zhang, 2018*; *Wang et al., 2019*). Pharmacological modulators of TRPM4 have been explored for a variety of conditions, including cancer, stroke, multiple sclerosis and heart disease (*Bianchi et al., 2018*; *Dienes et al., 2021*; *Kovács et al., 2022*).

We began by characterizing WT mouse TRPM4 (mTRPM4) using the inside-out configuration of the patch-clamp recording technique (*Hamill et al., 1981*) with $Na^+$ as the primary charge carrier in both intracellular and extracellular solutions. As described previously (*Launay et al., 2002*; *Nilius et al., 2003*; *Ullrich et al., 2005*; *Zhang et al., 2005*), we observed that mTRPM4 is activated by intracellular $Ca^{2+}$ in a concentration-dependent fashion and exhibits an outwardly rectifying current–voltage (*I–V*) relation in the presence of $Ca^{2+}$ (*Figure 3*). Partial rundown during the first $Ca^{2+}$ application in each patch was observed, consistent with activity-dependent PIP2 depletion previously reported (*Zhang et al., 2005*; *Nilius et al., 2006*; *Guo et al., 2017*), and the $Ca^{2+}$ sensitivity we measured after the first $Ca^{2+}$ application is similar to previously reported $EC_{50}$ values for TRPM4 in inside-out patches that have not been supplemented with PIP2 (*Zhang et al., 2005*; *Nilius et al., 2006*; *Guo et al., 2017*).

We first tested the sensitivity of mTRPM4 to icilin and found that the channel was not activated by 25 µM icilin applied alone over a wide range of voltages (*Figure 3A–C*). Icilin sensitivity in TRPM8 is known to require $Ca^{2+}$ as a co-agonist, so we next tested whether icilin affects the activation of mTRPM4 by different concentrations of $Ca^{2+}$. In contrast to our prediction that TRPM4 would be insensitive to icilin based on G805 in TRPM8 not being conserved in TRPM4 (A867), we observed that icilin potentiates the activation mTRPM4 by intracellular $Ca^{2+}$ (*Figure 3A–C*). This potentiation occurs at both positive and negative voltages at subsaturating $Ca^{2+}$ concentrations (500 µM). In contrast, at saturating concentrations of $Ca^{2+}$ (3 mM) (*Nilius et al., 2006*; *Guo et al., 2017*), minimal potentiation is observed at positive voltages, but notable potentiation is observed at negative voltages in both *I–V* and conductance–voltage (*G–V*) relations, indicating that icilin also diminishes the extent of outward-rectification (*Figure 3A–C*). Icilin also changes the kinetics of TRPM4 activation by voltage steps by enhancing the fraction of current elicited instantaneously after voltage steps and diminishing the fraction that activates more slowly on the timescale of 100–200 ms (*Figure 3A*). To quantify this, we defined steady-state currents ($I_{ss}$) observed at the end of 200 ms voltage steps to a maximally activating voltage of +160 mV as the sum of current that activates instantaneously upon depolarization ($I_{inst}$) and the relaxing current that slowly activates during the voltage step (*Figure 4A*). The fraction of instantaneous current ($I_{inst}/I_{ss}$) increased with both $Ca^{2+}$ and icilin (*Figure 4B*), indicating that both stimuli diminish outward-rectification. We also noticed that closure of TRPM4 channels following removal of both $Ca^{2+}$ and icilin appeared to be slower compared to when the channel was only activated by $Ca^{2+}$ alone (*Figure 4C, D*). From these results, we concluded that icilin interacts with TRPM4 and modulates activation of the channel by intracellular $Ca^{2+}$.

We also tested whether icilin modulates TRPM3 as several of the residues in TRPM8 that are critical for activation by cooling agents are not conserved in TRPM3. TRPM3 is activated by pregnenolone sulfate (PregS) and exhibits an outward-rectifying *I–V* relationship with that agonist (*Oberwinkler et al., 2005*; *Vriens et al., 2011*; *Held et al., 2015*), a property that is also common to both TRPM8 and TRPM4 in response to their activators. Because TRPM3 is inhibited by external $Na^+$ ions (*Oberwinkler et al., 2005*), similar to TRPV1 (*Jara-Oseguera et al., 2016*), we used $Cs^+$ as the primary charge carrier and recorded the activity of TRPM3 over a wide range of voltages before and after application of PregS. Application of 25 µM icilin did not appear to activate TRPM3 when applied alone and we observed outwardly rectifying currents in response to PregS application, but these were not detectably altered by the prior application of icilin (*Figure 4—figure supplement 1*).

## TRPM4 mutations altering the effects of icilin

We next tested whether the icilin sensitivity observed in TRPM4 is mediated by binding of the cooling agent to the site identified in TRPM8. We first identified cooling agent binding pocket residues where mutations have been shown to specifically affect icilin sensitivity in TRPM8 without disrupting $Ca^{2+}$ activation of TRPM4. Although some of the previously identified mutations like N799 and D802 in

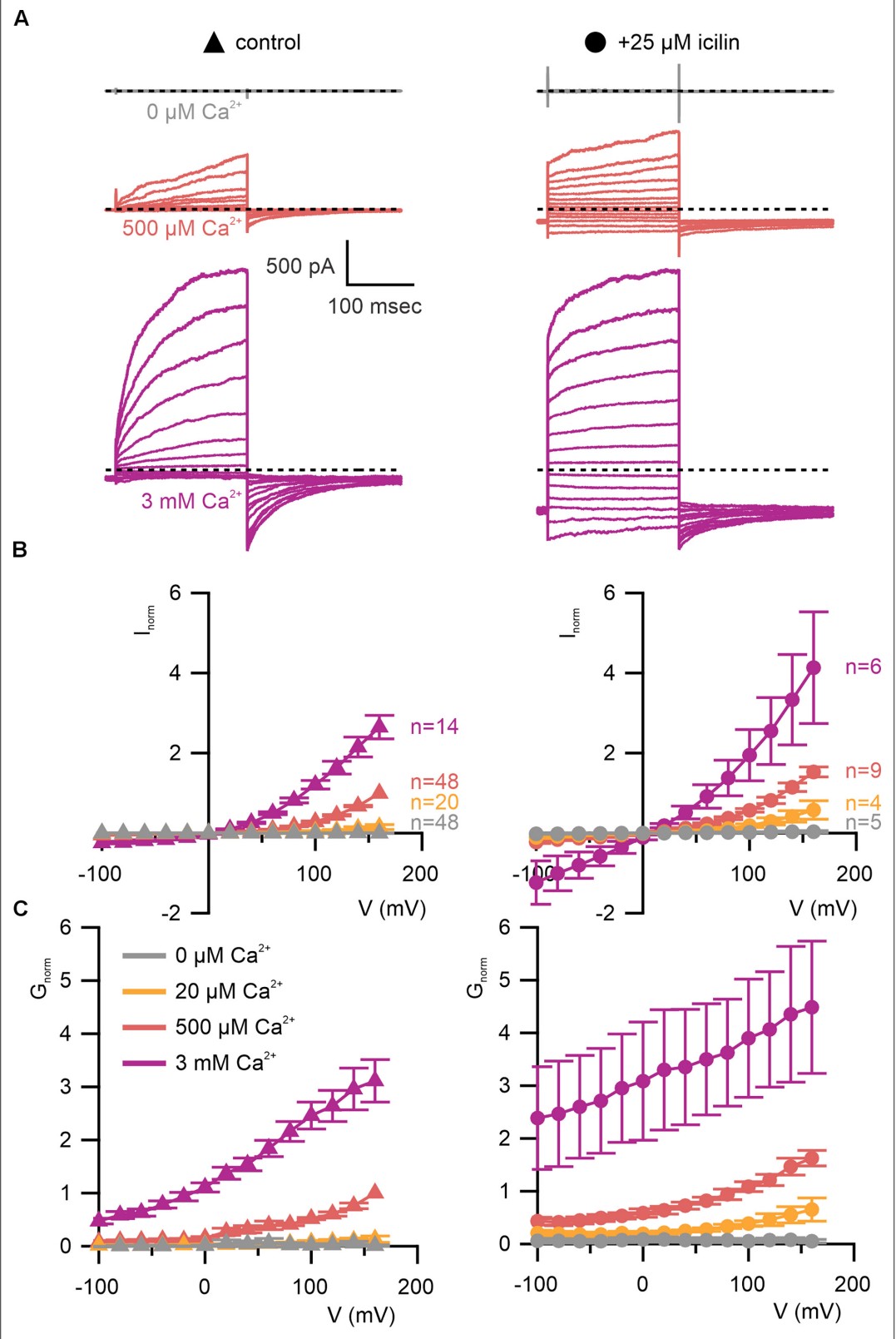

**Figure 3.** WT TRPM4 is sensitive to intracellular $Ca^{2+}$, voltage, and icilin. (**A**) Sample current families obtained using a holding voltage of −60 mV with 200 ms steps to voltages between −100 and +160 mV (Δ 20 mV) before returning to −60 mV. Control traces in the left column were obtained with TRPM4 in the absence of icilin and the presence of the labeled $Ca^{2+}$ concentrations, and traces in the right column were obtained in the presence of 25 µM icilin

*Figure 3 continued on next page*

*Figure 3 continued*

and the labeled Ca$^{2+}$ concentrations. (**B**) Normalized *I–V* and (**C**) normalized *G–V* plots for populations of cells in the absence (left, triangles) or presence (right, circles) of 25 µM icilin. Conductance values were obtained from tail current measurements. For each cell, values are normalized to the steady-state current or conductance at +160 mV in the presence of 500 µM Ca$^{2+}$. Error bars indicate standard error of the mean.

The online version of this article includes the following source data for figure 3:

**Source data 1.** Excel file with electrophysiology data for *Figure 3B*.

**Source data 2.** Excel file with electrophysiology data for *Figure 3C*.

rTRPM8 and hTRPM8 disrupt icilin sensitivity (*Chuang et al., 2004*; *Winking et al., 2012*; *Kühn et al., 2013*; *Beccari et al., 2017*), they likely do so by disrupting binding of the obligate cofactor Ca$^{2+}$ (*Yin et al., 2019a*; *Zhao et al., 2022*) and mutations in the equivalent Ca$^{2+}$-coordinating residues in rTRPM4 (N859, D862) have been shown to decrease Ca$^{2+}$ affinity (*Yamaguchi et al., 2019*). In contrast, one of the key determinants of icilin sensitivity in rTRPM8 is G805 (*Chuang et al., 2004*), which is directly adjacent to L806 within the icilin binding pocket even though G805 itself is not within 4 Å of icilin in the available structures (*Figure 2*). G805 in rTRPM8 corresponds to an Ala in icilin-insensitive chicken TRPM8 (cTRPM8), and the G805A mutation decreases the sensitivity of rTRPM8 to icilin while the inverse Ala to Gly mutation in cTRPM8 or faTRPM8 introduces sensitivity to icilin (*Chuang et al., 2004*; *Yin et al., 2019a*). We hypothesized that because mTRPM4 is already sensitive to icilin, making the equivalent A867G mutation might further enhance the sensitivity of the channel to icilin.

We first tested whether the A867G mutation in mTRPM4 alters channel activation and observed Ca$^{2+}$ sensitivity and outward-rectification (*Figure 5*) that was similar to WT (*Figure 3*). We then applied icilin and observed that, as observed in TRPM4 and TRPM8, icilin applied in the absence of Ca$^{2+}$ was not sufficient to robustly activate the A867G mutant (*Figure 5A–C*). When applied in the presence of intracellular Ca$^{2+}$, icilin potentiated Ca$^{2+}$-evoked currents even more dramatically in the A867G mutant than in the WT channel (*Figure 5*). Notably, maximal currents were observed by application of 500 µM Ca$^{2+}$ in the presence of icilin, which is a subsaturating concentration in the WT channel, even in the presence of icilin (*Figure 3*). Icilin also diminished the extent of outward-rectification dramatically in the A867G mutant channel, which was observed as particularly dramatic potentiation of currents elicited at negative voltages compared to positive voltages and a nearly linear *I–V* relation at a saturating concentration of Ca$^{2+}$ (*Figure 5*). This loss of outward-rectification could also be observed by quantifying the fraction of current activated instantaneously, where A867G mutant channels showed Ca$^{2+}$-dependent increases in $I_{inst}/I_{ss}$ similar to WT in the absence of icilin, but $I_{inst}/I_{ss}$ values near 1 in the presence of 25 µM icilin (*Figure 6*). The A867G mutation does not modulate TRPM4 sensitivity to Ca$^{2+}$ applied alone, indicating that this mutation specifically altered modulation by icilin. The selective influence of the A867G mutation on icilin may be related to the observation in TRPM8 where the G805A mutation affects icilin sensitivity without altering sensitivity to menthol (*Chuang et al., 2004*). We also noticed that closure of TRPM4 following removal of both Ca$^{2+}$ and icilin was considerably slower in the A867G mutant compared to the WT channel (*Figure 6C, D*), suggesting that the mutant binds icilin with higher affinity than the WT channel and that the slowing of channel closure is due to slower icilin unbinding (*Figure 4C, D*). Taken together, the effects of the A867G mutation are consistent with the icilin sensitivity of TRPM4 being mediated by the equivalent binding pocket as in TRPM8.

Another critical determinant of icilin sensitivity in hTRPM8 is R842 (*Figure 2*), a residue where mutations to His dramatically diminish activation by the cooling agents icilin and menthol and that is within 4 Å of key substituent groups in icilin in the two possible docking orientations of the cooling agent (*Voets et al., 2007*; *Palchevskyi et al., 2023*). If icilin binds to the equivalent pocket in TRPM4 as in TRPM8, we hypothesized that the equivalent mutation in TRPM4 (R901H) would disrupt icilin sensitivity. As with the A867G mutant, we began by testing whether R901H in TRPM4 exhibits similar behavior to the WT channel and observed Ca$^{2+}$ sensitivity similar to WT TRPM4, though it notably enhanced outward-rectification (*Figure 7*). The R901H mutant was insensitive to icilin applied alone, as observed for WT and the A867G mutant in mTRPM4, however, unlike WT and the A867G mutant in mTRPM4, activation of the R901H mutant is not enhanced by icilin, regardless of whether subsaturating or saturating concentrations of intracellular Ca$^{2+}$ are tested (*Figure 7*), and icilin does not alter the instantaneously activating fraction of current or prolong channel closure (*Figure 8*). These results in the R901H mutant of TRPM4 are consistent with the loss of icilin sensitivity in R842H mutant of

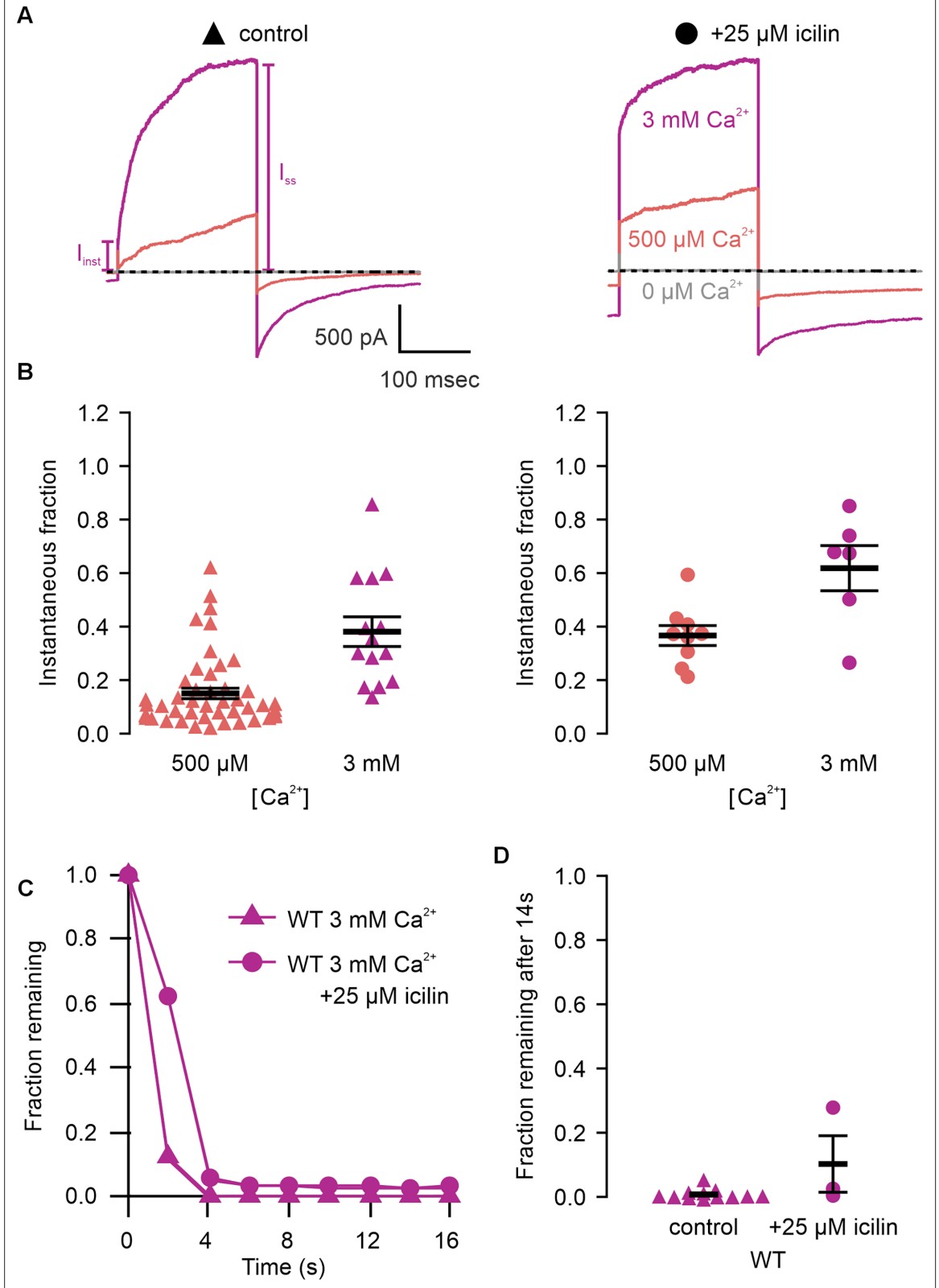

**Figure 4.** Icilin modulates voltage-dependent activation and closure kinetics of TRPM4. (**A**) Sample current traces illustrating the fraction of current that activates rapidly ($I_{inst}$) compared to the steady-state current at the end of the pulse ($I_{SS}$). The pulse protocols used a holding voltage of −60 mV with 200 ms steps to +160 mV in the presence of varying concentrations of intracellular $Ca^{2+}$. Traces were obtained in the absence (left) or presence (right) of 25 μM icilin. (**B**) Instantaneous fraction of current ($I_{inst}/I_{SS}$) calculated using voltage steps to +160 mV at various concentrations of intracellular $Ca^{2+}$ for

*Figure 4 continued on next page*

*Figure 4 continued*

individual cells in the absence (left, triangles) or presence (right, circles) of 25 µM icilin. Error bars indicate standard error of the mean. (**C**) Fraction of current remaining after application of 3 mM $Ca^{2+}$ alone (triangles) or both 3 mM $Ca^{2+}$ and 25 µM icilin (squares) for WT TRPM4. Currents were elicited by voltage steps from −100 to +100 mV. (**D**) Fraction of current remaining 14 s after removal of 3 mM $Ca^{2+}$ alone (triangles) or both 3 mM $Ca^{2+}$ and 25 µM icilin (squares) for WT TRPM4. Currents were elicited by voltage steps from −100 to +160 mV.

The online version of this article includes the following source data and figure supplement(s) for figure 4:

**Source data 1.** Excel file with electrophysiology data for *Figure 4B–D*.

**Figure supplement 1.** Icilin does not modulate voltage-dependent activation of TRPM3α2.

**Figure supplement 1—source data 1.** Excel file with electrophysiology data for *Figure 4—figure supplement 1A*.

**Figure supplement 1—source data 2.** Excel file with electrophysiology data for *Figure 4—figure supplement 1B, C*.

rTRPM8, providing further evidence that icilin binds to the conserved binding pocket in hTRPM4 to modulate channel activation by intracellular $Ca^{2+}$.

## Discussion

The objective of the present study was to explore the conservation of the cooling agent binding pocket within TRPM channels. This site has been carefully interrogated in TRPM8, where structures are available with different cooling agents bound and where extensive mutagenesis has determined the influence of key residues in activation of the channel by cooling agents (*Chuang et al., 2004*; *Bandell et al., 2006*; *Voets et al., 2007*; *Malkia et al., 2009*; *Winking et al., 2012*; *Kühn et al., 2013*; *Beccari et al., 2017*; *Yin et al., 2018*; *Diver et al., 2019*; *Yin et al., 2019a*; *Yin and Lee, 2020*; *Plaza-Cayón et al., 2022*; *Zhao et al., 2022*). In TRPM8, the cooling agent binding pocket is contained within the S1–S4 domain and opens into the intracellular side of the membrane where a $Ca^{2+}$-ion-binding site is located. Our analysis across the available TRPM structures shows that the cooling agent binding pocket in TRPM8 is well-conserved in TRPM2, TRPM4 and TRPM5, but not in TRPM3 or TRPM7 (*Figure 2*). Using functional approaches, we explored the interaction of icilin with TRPM4 and found that the channel can be modulated by icilin (*Figures 3 and 4*). Two hallmark features of TRPM4 are that it is activated by intracellular $Ca^{2+}$ and that its *I–V* relation is outwardly rectifying, permeating cations out of the cell considerably more favorably than into the cell (*Figures 3, 5 and 7*). We found that icilin enhanced the sensitivity of the channel to intracellular $Ca^{2+}$ and reduced the extent of outward-rectification such that the channel conducts cations into the cell considerably more favorably at negative membrane voltages (*Figures 3 and 4*). Mutations established to promote or disrupt the actions of icilin in TRPM8 had similar effects in TRPM4 (*Figures 5–8*), leading us to conclude that icilin is likely binding to a conserved cooling agent binding pocket in both channels. Our analysis of TRPM3 channels revealed that key residues within the cooling agent binding pocket of TRPM8 are not conserved in TRPM3 (*Figure 2*) and our results also show that icilin does not detectably alter activation by PregS (*Figure 4—figure supplement 1*), providing a negative control for our findings with TRPM4. Collectively, these findings demonstrate that not only is the cooling agent binding pocket conserved between TRPM8 and TRPM4, but that coupling between occupancy of that pocket and opening of the channel are also conserved in both TRPM channels. The conserved nature of the coupling mechanism is particularly interesting when one considers that although the cooling agent binding pocket is highly conserved (89% identity), the S1–S6 regions exhibit much lower conservation (34% identity).

The identification of a conserved S1–S4 binding pocket in TRPM4 and TRPM8 raises many questions that would be interesting to interrogate in future studies. For example, there are many structurally distinct cooling agents that have been developed as agonists of TRPM8, some of which require $Ca^{2+}$ as a co-agonist and some that do not, and there are many antagonists thought to bind within the cooling agent binding pocket in TRPM8 (*Sherkheli et al., 2010*; *De Petrocellis et al., 2015*; *Diver et al., 2019*; *González-Muñiz et al., 2019*; *Yin et al., 2019a*; *Yin and Lee, 2020*; *Plaza-Cayón et al., 2022*; *Yin et al., 2022*; *Zhao et al., 2022*; *Yin et al., 2024*). It would be interesting to further explore the extent to which these ligands discriminate between the two channels and to try and solve structures of TRPM4 with some of these ligands bound to determine the extent to which binding poses are similar. Our analysis of the cooling agent binding pocket in TRPM channels suggests that both TRPM2 and TRPM5 might also be sensitive to icilin or related compounds because most of the key residues

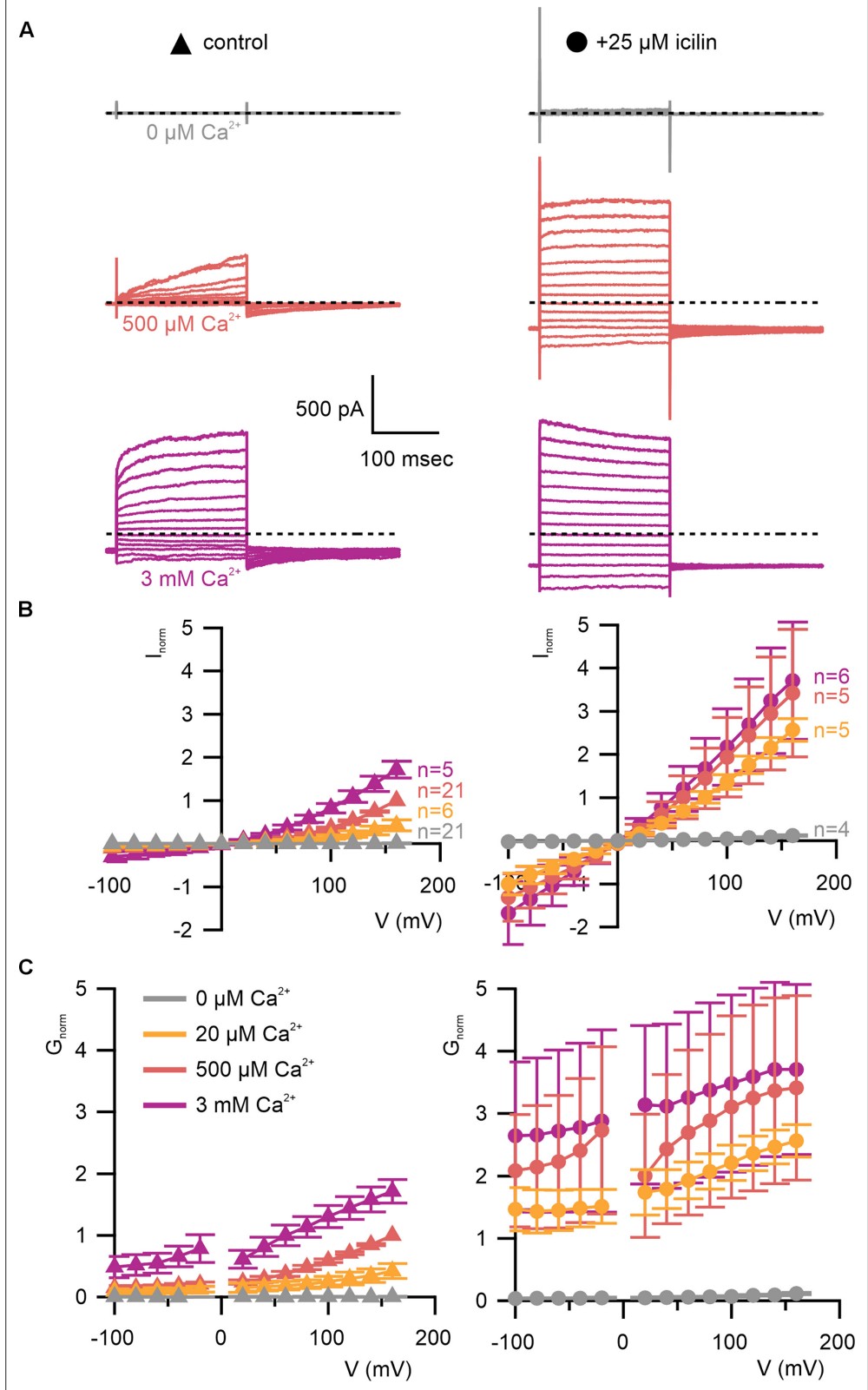

**Figure 5.** A867G mutant TRPM4 retains sensitivity to $Ca^{2+}$ and voltage, but has enhanced sensitivity to icilin. (**A**) Sample current families obtained using a holding voltage of −60 mV with 200 ms steps to voltages between −100 and +160 mV (Δ 20 mV) before returning to −60 mV. Control traces in the left column were obtained with A867G mTRPM4 in the absence of icilin and the presence of the labeled $Ca^{2+}$ concentrations, and traces in the right

*Figure 5 continued on next page*

*Figure 5 continued*

column were obtained in the presence of 25 μM icilin and the labeled $Ca^{2+}$ concentrations. (**B**) Normalized *I–V* and (**C**) normalized *G–V* plots for populations of cells in the absence (left, triangles) or presence (right, circles) of 25 μM icilin. Conductance values were calculated from steady-state currents. For each cell, values are normalized to the steady-state current or conductance at +160 mV in the presence of 500 μM $Ca^{2+}$. Error bars indicate standard error of the mean.

The online version of this article includes the following source data for figure 5:

**Source data 1.** Excel file with electrophysiology data for *Figure 5B*.

**Source data 2.** Excel file with electrophysiology data for *Figure 5C*.

are conserved, and these would be interesting directions to explore. Also, our experiments were done after depletion of PIP2 with the first intracellular $Ca^{2+}$ application, which leads to partial rundown of the channel and a pronounced lowering of $Ca^{2+}$ affinity (*Zhang et al., 2005*; *Nilius et al., 2006*; *Guo et al., 2017*). It would be fascinating to investigate the extent to which PIP2 modulates the action of icilin on TRPM4. For example, the dramatic influence of PIP2 on $Ca^{2+}$ sensitivity might enhance the affinity of icilin, which we only studied at a relatively high concentration that is saturating for TRPM8 (*Andersson et al., 2004*; *Chuang et al., 2004*).

The conservation of cooling agent binding pockets in TRPM2, TRPM4, and TRPM5, and the finding that icilin can modulate TRPM4 is interesting because two other TRP channels have been reported to be sensitive to icilin even though our analysis shows that the cooling agent binding pocket in TRPM8 is not conserved in those channels. The first example is TRPA1, which can be activated by icilin even though the residues in TRPA1 correspond to those that form the cooling agent binding pocket in TRPM8 are poorly conserved (*Figure 2*). As observed for TRPM8, TRPA1 icilin sensitivity has been reported to require coactivation with $Ca^{2+}$ (*Doerner et al., 2007*), and the S1–S4 $Ca^{2+}$-binding site observed in TRPMs is indeed conserved in TRPA1, though the S2 $Ca^{2+}$-coordinating residues are located one helical turn lower in TRPA1 when its structures are aligned with TRPM8 (*Figure 2*). $Ca^{2+}$ acts directly to activate and inactivate TRPA1 at low and high concentrations, respectively (*Wang et al., 2008*; *Hasan and Zhang, 2018*), and can also act indirectly on TRPA1 through $Ca^{2+}$-binding proteins such as calmodulin (*Hasan et al., 2017*). Mutations in this TRPA1 TM $Ca^{2+}$-binding site do affect $Ca^{2+}$ potentiation and desensitization (*Zimova et al., 2018*; *Zhao et al., 2020*). However, it is unclear whether this conserved TM $Ca^{2+}$-binding site directly mediates icilin sensitivity in TRPA1 as it does in TRPM8 because multiple other $Ca^{2+}$-binding sites have been proposed in TRPA1, including EF hand motifs near the N-terminus (*Doerner et al., 2007*; *Zurborg et al., 2007*), residues within the TM helices S2 and S3 (*Zhao et al., 2020*), and acidic residues at the C-terminus (*Sura et al., 2012*). Overall, the lack of conservation of the TRPM8 cooling agent binding pocket in TRPA1 raises the possibility that icilin binds to a different location to regulate the activity of that TRP channel.

A second example is TRPV3, which can be inhibited by icilin in a calcium-dependent manner (*Sherkheli et al., 2012*; *Billen et al., 2015*). It is unclear where icilin binds to TRPV3, though tryptophan fluorescence quenching experiments on mutants have revealed that residues W481 at the top of S2, W559 in the middle of S4, and W710 after the TRP box in TRPV3 are not directly involved in icilin binding (*Billen et al., 2015*). The residues equivalent to the icilin binding pocket in TRPM8 are not conserved in TRPV3 and the TM $Ca^{2+}$ ion binding site observed in TRPM channels and TRPA1 is also not conserved in TRPV3 (*Figure 2*), suggesting that the observed $Ca^{2+}$ dependence of the icilin sensitivity in TRPV3 is mediated by a different part of the channel, possibly through direct binding of $Ca^{2+}$ ions or indirectly through calmodulin binding (*Xiao et al., 2008*; *Phelps et al., 2010*). An arginine in the TRP box and an aspartate in the pore loop have been implicated in the $Ca^{2+}$ dependence of TRPV3 and other TRPV channels (*García-Martínez et al., 2000*; *Voets et al., 2002*; *Chung et al., 2004*; *Xiao et al., 2008*; *Hu et al., 2009*; *Xiao et al., 2008*), so these would be potential sites to explore. As in TRPA1, the lack of conservation of the cooling agent binding pocket found in TRPM8 with the corresponding region of TRPV3 raises the possibility that icilin and $Ca^{2+}$ bind to different regions in TRPV3 compared to TRPM channels.

The discovery that the cooling agent icilin can interact with TRPM4 and modulate its activation by $Ca^{2+}$ and voltage has several important implications for understanding key mechanisms in TRP channels. First, it provides another clear example of two TRP channels that are activated by very different stimuli, cooling agents and noxious cold in the case of TRPM8 and intracellular $Ca^{2+}$ in the case of

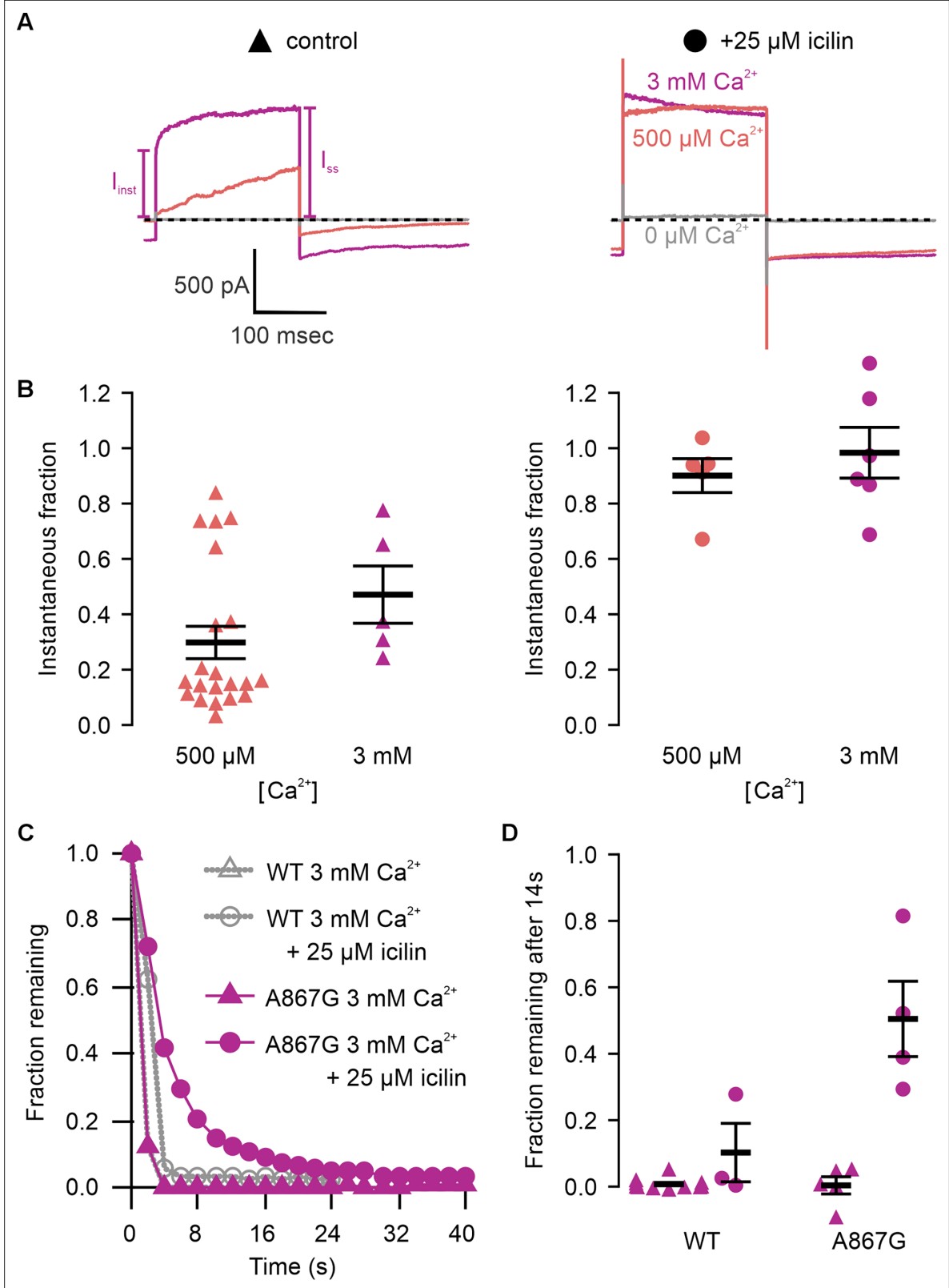

**Figure 6.** Icilin modulation of TRPM4 is enhanced in the A867G mutant. (**A**) Sample current traces illustrating the fraction of current that activates rapidly ($I_{inst}$) compared to the steady-state current at the end of the pulse ($I_{SS}$). The pulse protocols used a holding voltage of −60 mV with 200 ms steps to +160 mV in the presence of varying concentrations of intracellular $Ca^{2+}$. Traces were obtained in the absence (left) or presence (right) of 25 μM icilin. (**B**) Instantaneous fraction of current ($I_{inst}/I_{SS}$) calculated using +160 mV voltage steps at various concentrations of intracellular $Ca^{2+}$ for individual cells in

*Figure 6 continued on next page*

Figure 6 continued

the absence (left, triangles) or presence (right, circles) of 25 μM icilin. Error bars indicate standard error of the mean. (**C**) Fraction of current remaining after application of 3 mM $Ca^{2+}$ alone (triangles) or both 3 mM $Ca^{2+}$ and 25 μM icilin (squares) for WT TRPM4 (gray) or A867G TRPM4 (purple). Currents were elicited by voltage steps from −100 to +100 mV. (**D**) Fraction of current remaining 14 s after removal of 3 mM $Ca^{2+}$ alone (triangles) or both 3 mM $Ca^{2+}$ and 25 μM icilin (squares) for WT (left) or A867G TRPM4 (right). Currents were elicited by voltage steps from −100 to +160 mV.

The online version of this article includes the following source data for figure 6:

**Source data 1.** Excel file with electrophysiology data for *Figure 6B–D*.

TRPM4, but that share common underlying mechanisms wherein binding sites for ligands within the S1–S4 domain are coupled to channel opening. Conceptually, our findings with TRPM4 relate to those in TRPV channels, where sensitivity to activation by vanilloids can be engineered into the vanilloid-insensitive TRPV2 and TRPV3 channels with relatively few mutations within the pocket corresponding to where vanilloids bind in TRPV1 (*Yang et al., 2016*; *Zhang et al., 2016*; *Zubcevic et al., 2018*; *Zhang et al., 2019*). In the case of TRPV2 and TRPV3, the ability to open in response to occupancy of the vanilloid binding site is intrinsic to those channels and requires only sculpting of the vanilloid site to enable ligand binding. Although we imagined that we might need to undertake similar engineering of TRPM4 to make it sensitive to cooling agents given the presence of an Ala at 867, we instead discovered intrinsic sensitivity to icilin that could be further tuned by the A867G mutation. It is fascinating that coupling between two distinct types of ligand binding pockets and channel opening are conserved in these examples of TRPV and TRPM channels, raising the possibility that key mechanisms of channel gating will turn out to be shared features of other TRP channels.

Second, uncovering a relationship between TRPM4 and TRPM8 provides motivation to explore the mechanistic relationships between TRPM channels further because their functional properties can vary quite dramatically, from being heat activated in TRPM3 (*Vriens et al., 2011*; *Vandewauw et al., 2018*) to cold activated in TRPM8 (*McKemy et al., 2002*; *Peier et al., 2002*; *Bautista et al., 2007*; *Dhaka et al., 2007*). TRPM channels also exhibit varying rectification properties, for example, with TRPM4 exhibiting pronounced outward-rectification at all $Ca^{2+}$ concentrations (*Launay et al., 2002*; *Nilius et al., 2003*; *Ullrich et al., 2005*; *Zhang et al., 2005*) and TRPM5 exhibiting outward-rectification at low $Ca^{2+}$ concentrations and a nearly linear *I–V* relationship at high concentrations (*Liu and Liman, 2003*; *Yamaguchi et al., 2019*; *Ruan et al., 2021*). Here, we show that TRPM4 can exhibit these two extremes of rectification, from outwardly rectifying in response to $Ca^{2+}$ alone to a linear *I–V* in the presence of $Ca^{2+}$ and icilin (*Figures 3 and 5*). Related differences in rectification in TRPM3 when activated by different agonists have factored into proposals for distinct ion permeation pathways; a conventional pore at the central axis and second within the S1–S4 domains (*Vriens et al., 2014*). Our observation that cooling agents can modulate the rectifying properties of TRPM4 quite dramatically may be mechanistically related to observations in TRPM3, but at least in TRPM4 it is hard to imagine how ions could conceivably permeate through the S1–S4 domain with icilin bound (*Figures 1 and 2*). A more likely explanation is that cooling agent binding to TRPM4 not only couples to channel opening, but also to a voltage-dependent mechanism that underlies rectification. It will be fascinating to explore how $Ca^{2+}$ and cooling agent binding in such close proximity within the S1–S4 domain of TRPM4 can have such divergent effects on the mechanism of rectification.

Third, the discovery that cooling agents can bind to and modulate TRPM4 raises intriguing questions about why this site is conserved in so many TRPM channels. One possible answer is that there are endogenous, yet to be identified molecules that can bind to the cooling agent binding site to regulate the activity of TRPM channels. Conceptually, a role of endogenous modulators binding to the cooling agent binding pocket is similar to lipids occupying the vanilloid binding site in TRPV channels to stabilize the closed state of the channel, which are displaced when activators bind to activate the channel (*Cao et al., 2013a*; *Gao et al., 2016*; *Nadezhdin et al., 2021*; *Zhang et al., 2021*; *Arnold et al., 2024*; *Nadezhdin et al., 2024*).

Finally, our findings here with TRPM4 may also be relevant for understanding the neuroprotective actions of icilin in mouse models of multiple sclerosis and seizures (*Pezzoli et al., 2014*; *Ewanchuk et al., 2018*; *Moriyama et al., 2019*). Some of these neuroprotective effects remain in TRPM8 or TRPA1 knockout animals, raising the intriguing possibility that TRPM4 may underlie the actions of icilin in these disease models. Indeed, TRPM4 inhibitors and TRPM4 knockouts have been previously

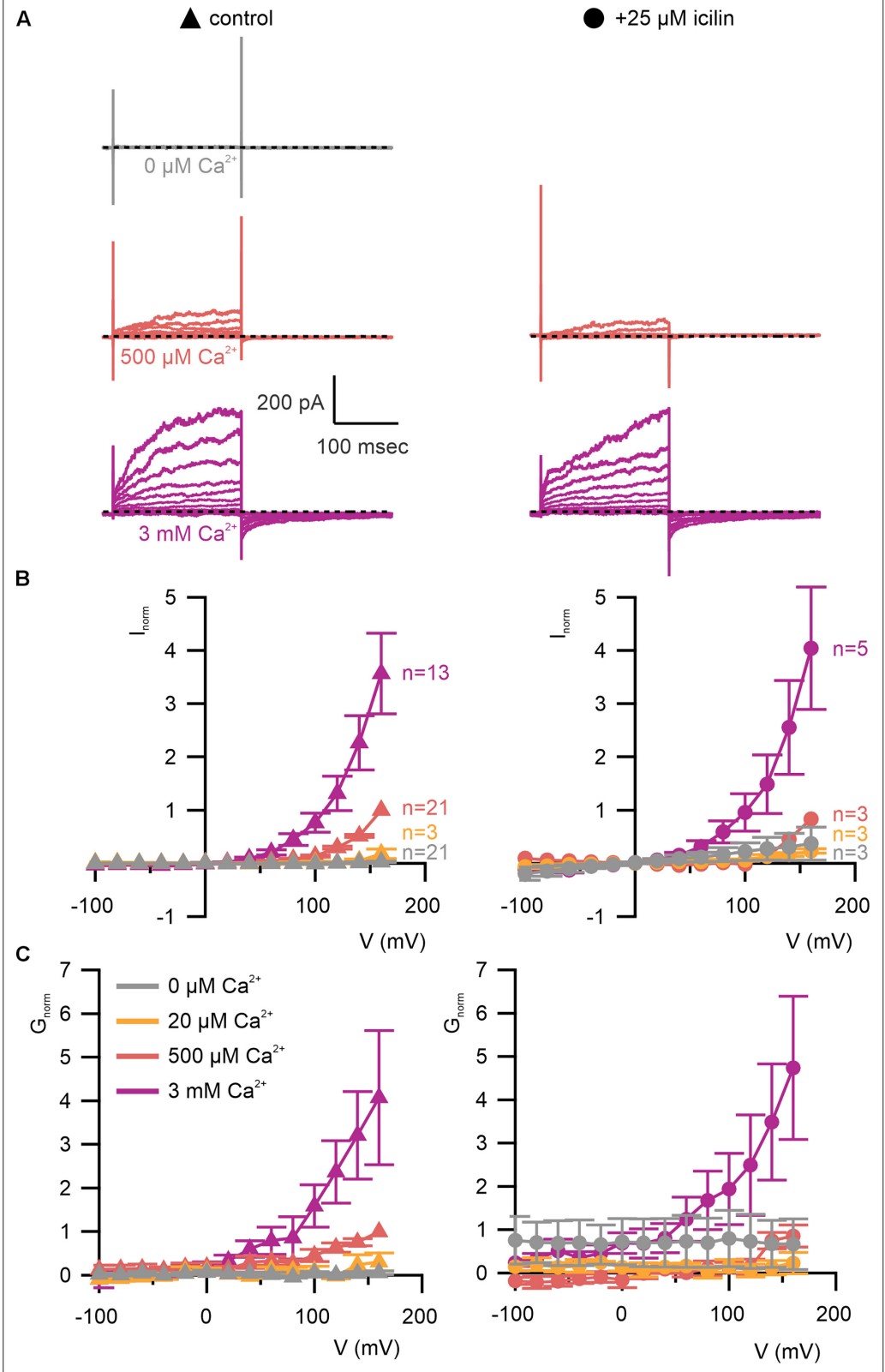

**Figure 7.** R901H mutant TRPM4 is sensitive to intracellular $Ca^{2+}$ and voltage, but icilin does not promote opening. (**A**) Sample current families obtained using a holding voltage of −60 mV with 200 ms steps to voltages between −100 and +160 mV (Δ 20 mV) before returning to −60 mV. Control traces in the left column were obtained with R901H mTRPM4 in the absence of icilin and the presence of the labeled $Ca^{2+}$ concentrations, and traces in the

*Figure 7 continued on next page*

*Figure 7 continued*

right column were obtained in the presence of 25 µM icilin and the labeled $Ca^{2+}$ concentrations. For the cell shown, current families were not obtained in the presence of icilin and the absence of $Ca^{2+}$. (**B**) Normalized *I–V* and (**C**) normalized *G–V* plots for populations of cells in the absence (left, triangles) or presence (right, circles) of 25 µM icilin. Conductance values were obtained from tail current measurements. For each cell, values are normalized to the steady-state current or conductance at +160 mV in the presence of 500 µM $Ca^{2+}$. Error bars indicate standard error of the mean.

The online version of this article includes the following source data for figure 7:

**Source data 1.** Excel file with electrophysiology data for *Figure 7B*.

**Source data 2.** Excel file with electrophysiology data for *Figure 7C*.

described to play a role in these same mouse models of multiple sclerosis, inviting further investigation of TRPM4 modulators in this disease (*Bianchi et al., 2018*).

# Materials and methods

## Key resources table

| Reagent type (species) or resource | Designation | Source or reference | Identifiers | Additional information |
|---|---|---|---|---|
| Cell line (*Homo sapiens*) | HEK293 | ATCC | CRL-1573 ATCC Cat# PTA-4488, RRID:CVCL_0045 | |
| Transfected construct (*Mus musculus*) | mTRPM4b (plasmid) | Received from Youxing Jiang | | pEGFP-N1 vector |
| Transfected construct (*Mus musculus*) | mTRPM3α2 (plasmid) | Received from Thomas Voets | | pCAGGS/IRES-GFP vector |
| Commercial assay or kit | FuGENE6 Transfection Reagent | Promega | E269 | |
| Commercial assay or kit | QuikChange Lightning | Agilent Technologies | 21051 | |
| Chemical compound, drug | Icilin | Sigma | CAS 36945-98-9 | Source # 0000088284 Source # 0000141718 |
| Chemical compound, drug | Pregnenolone sulfate | Sigma | P162-25MG | Lot # MKCQ8813 |
| Software, algorithm | IgorPro | WaveMetrics | RRID:SCR_000325 | Data analysis |
| Software, algorithm | Python programming language | | RRID:SCR_008394 | Data analysis |
| Software, algorithm | SciPy | | RRID:SCR_008058 | Data analysis |
| Software, algorithm | Matplotlib | | RRID:SCR_008624 | Data visualization |
| Software, algorithm | seaborn | | RRID:SCR_018132 | Data visualization |
| Software, algorithm | PyMOL | Schrödinger | RRID:SCR_000305 | Structure visualization |
| Software, algorithm | PoseFilter | Subha Kalyaanamoorthy; https://doi.org/10.1093/bioinformatics/btab188 | | Data analysis PyMOL Plugin |
| Software, algorithm | pClamp 11 | Molecular Devices | RRID:SCR_011323 | Acquisition patch clamp data |
| Software, algorithm | JalView | | RRID:SCR_006459 | Sequence alignment visualization |
| Software, algorithm | Fr-TM-Align | Pandit & Skolnick; https://doi.org/10.1186/1471-2105-9-531 | | Data analysis |

## Structure-based sequence alignment

Structural alignment of TRPM and TRPA1 structure TM domains (pre-S1-TRP box) was performed using Fr-TM-Align as described previously (*Huffer et al., 2020*). Representative structures were selected based on nominal resolution. Sequence alignments were made in Jalview. Structures were visualized with PyMOL.

## Channel constructs

The mTRPM4b DNA plasmid tagged with EGFP at the C-terminus was a generous gift from Dr. Youxing Jiang (UT Southwestern) and was transfected into HEK293 cells using FuGENE6 transfection reagent. Mouse TRPM3α2 DNA in the bicistronic pCAGGS/IRES-GFP vector (*Vriens et al., 2014*) was provided by Dr. Thomas Voets (Catholic University, Leuven, Belgium) and transfected as for mTRPM4.

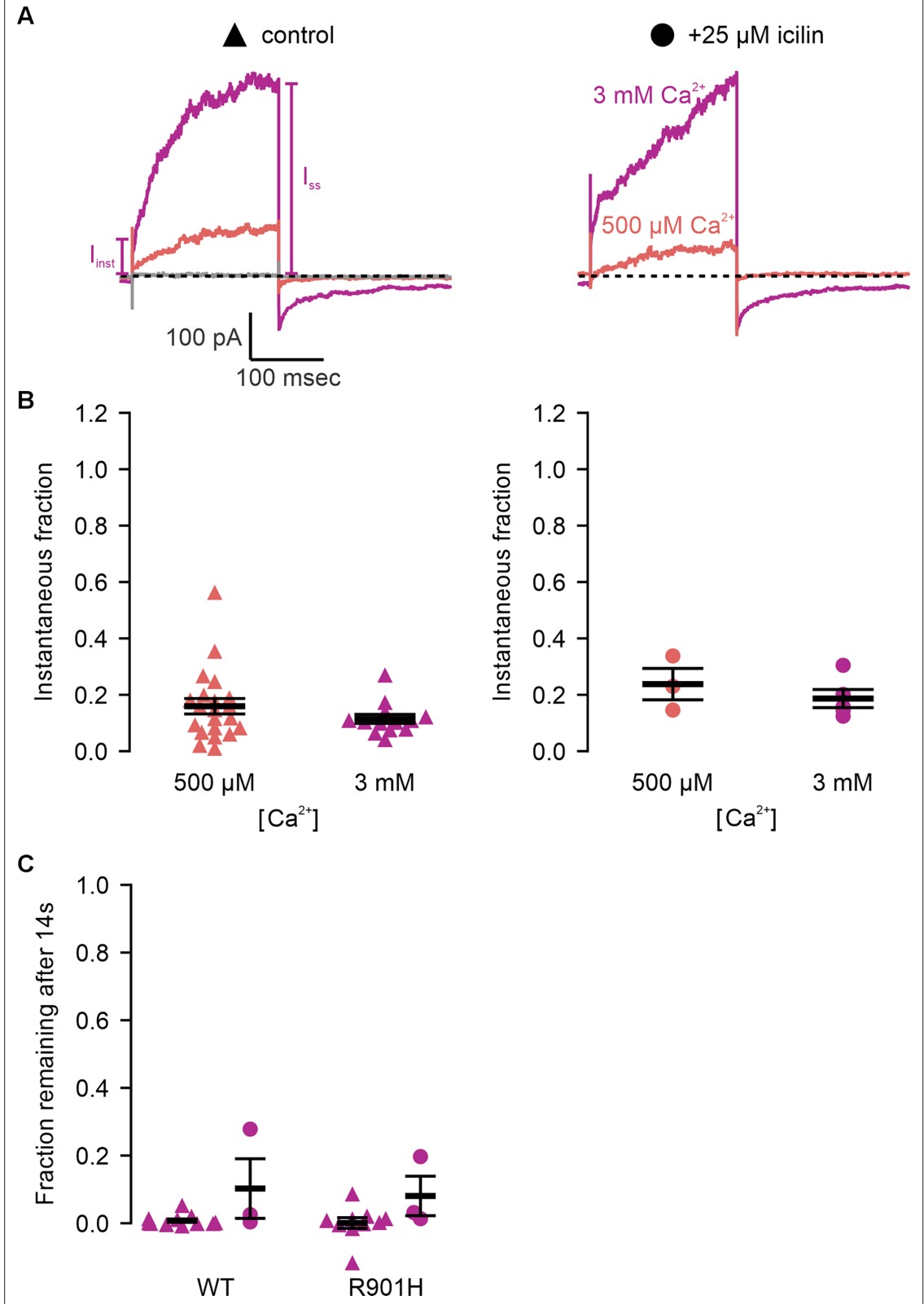

**Figure 8.** Icilin modulation of the voltage-dependent activation of TRPM4 is disrupted in the R901H mutant. (**A**) Sample current traces illustrating the fraction of current that activates rapidly ($I_{inst}$) compared to the steady-state current at the end of the pulse ($I_{ss}$). The pulse protocols used a holding voltage of −60 mV with 200 ms steps to +160 mV in the presence of varying concentrations of intracellular $Ca^{2+}$. Traces were obtained in the absence (left) or presence (right) of 25 µM icilin. For the cell shown, current families were not obtained in the presence of icilin and absence of $Ca^{2+}$. (**B**)

*Figure 8 continued on next page*

*Figure 8 continued*

Instantaneous fraction of current ($I_{inst}/I_{SS}$) calculated using +160 mV voltage steps at various concentrations of intracellular $Ca^{2+}$ for individual cells in the absence (left, triangles) or presence (right, circles) of 25 µM icilin. Error bars indicate standard error of the mean. (**C**) Fraction of current remaining after 14 s of 0 mM $Ca^{2+}$ wash, following removal of 3 mM $Ca^{2+}$ alone (triangles) or both 3 mM $Ca^{2+}$ and 25 µM icilin (squares) for WT (left) or R901H TRPM4 (right). Currents were measured between −100 and +160 mV (Δ20 mV), but only +160 mV current fractions are shown.

The online version of this article includes the following source data for figure 8:

**Source data 1.** Excel file with electrophysiology data for *Figure 8B, C*.

All mutations in mTRPM4 were made using the QuikChange Lightning technique (Agilent Technologies) and confirmed by DNA sequencing (Macrogen).

## Cell culture

Authenticated Human Embryonic Kidney (HEK293) cells were obtained from ATCC (CRL-1573) and cultured in Dulbecco's modified Eagle's medium supplemented with 10% fetal bovine serum and 10 mg l⁻¹ of gentamicin. HEK293 cells between passage numbers 5–25 were used and passaged when cells were between 40% and 80% confluent. Mycoplasma contamination was routinely tested and found to be negative. The cells were treated with trypsin and then seeded on glass coverslips at about 15% of the original confluency in 35 mm Petri dishes. Transfections were done using the FuGENE6 Transfection Reagent (Promega). Transfected cells were incubated at 37°C with 95% air and 5% $CO_2$ overnight for use in patch-clamp recordings; 16–48 hr depending on the construct.

## Electrophysiology

For recording the activity of TRPM4 in inside-out patches, the pipette (extracellular) solution contained 130 mM NaCl, 2 mM $MgCl_2$, 0.5 mM $CaCl_2$, 10 mM HEPES (4-(2-Hydroxyethyl)piperazine-1-ethane-sulfonic acid), with pH adjusted to 7.4 using NaOH. The bath (intracellular) solutions all contained 125 mM CsCl, 5 mM NaCl, 2 mM $MgCl_2$, 10 mM HEPES, with pH adjusted to 7.4 using CsOH. $CaCl_2$ was varied, and 0 mM $CaCl_2$ solutions contained 1 mM EGTA (ethylene glycol tetraacetic acid). For EGTA-containing solutions, $MgCl_2$ concentrations were adjusted using MaxChelator such that there was 2 mM free $MgCl_2$ (***Bers et al., 2010***). For recording the activity of TRPM3 in whole-cell recordings, the pipette (intracellular) solution contained 130 mM CsCl, 1 mM (free) $MgCl_2$, 10 mM HEPES, 10 mM EGTA, with pH adjusted to 7.2 using CsOH. The external (extracellular) solution contained 130 mM CsCl, 1 mM $MgCl_2$, 10 mM HEPES, with pH adjusted to 7.4 using CsOH. The bathing solution in which seals were obtained contained 130 mM NaCl, 1 mM $MgCl_2$, 10 mM HEPES, with pH adjusted to 7.4 with NaOH. Bath and ground chambers were connected by an agar bridge containing 3 M KCl. Icilin powder (Sigma) was dissolved in DMSO to a stock concentration of 40 mM, aliquoted, and stored at −80°C. As icilin is known to degrade (***Kühn et al., 2009***), the stock was not subjected to repeated freeze–thaw cycles. $Cs^+$-based solutions increased successful Giga seal formation over Na-based solutions. $K^+$ was omitted to prevent contamination of Kv channel currents. $Mg^{2+}$ was added to all solutions to inhibit endogenous TRPM7 currents that were observed in some passages (***Nadler et al., 2001***; ***Hermosura et al., 2002***; ***Kozak and Cahalan, 2003***). Patch pipette resistance ranged from 1 to 5 MOhms, with a typical value around 3 MOhms. Inside-out patch-clamp recordings were performed using a holding voltage of −60 mV with 200 ms steps to voltages between −100 and +160 mV (Δ20 mV). Electrophysiology data was acquired with an Axopatch 200B amplifier at a sampling frequency of 10 kHz and filtered to 5 kHz with a low-pass filter. Variable PIP2-depletion-induced current rundown was observed (***Nilius et al., 2006***), so all measurements were taken after current had reached a steady state. Patches that did not exhibit response to $Ca^{2+}$ were excluded because this either indicated that the pulled patch had formed a vesicle, precluding access to the intracellular face of the membrane, or that TRPM4 expression was too low for our experiments. For long timecourses with multiple exposures, only currents that returned to baseline upon removal of $Ca^{2+}$ were considered. Because icilin is thought to partition into the membrane, all non-icilin traces included in population data for *I–V* and *G–V* relations were collected prior to the application of icilin, even if icilin appeared to wash during the experiment. Patches that exhibited very large currents (>5 nA) were also excluded because they would result in substantial voltage errors and/or changes in the concentrations of ions. Currents were normalized to steady-state currents obtained in the presence of 500 µM $Ca^{2+}$ at +160 mV for each

cell. Baseline current subtraction was performed for each cell by subtracting leak currents obtained in the absence of $Ca^{2+}$.

Statistical methods were not used to determine the sample size. Sample size for electrophysiological studies was determined empirically by comparing individual measurements with population data obtained under differing conditions until convincing differences or lack thereof were evident. Additional exploratory experiments performed to determine ideal recording conditions are consistent with the data reported here, but these pilot data are not included in our analysis due to changes in experimental conditions, including varying voltage pulse protocols and different solution compositions. For all electrophysiological experiments, $n$ values represent the number of patches studied from between 9 and 31 different batches of cells. Electrophysiology data analysis and visualization were performed with IgorPro and Python (matplotlib).

## Acknowledgements

We thank Andres Jara-Oseguera, Surbhi Dhingra, Ana I Fernández-Mariño, Xiao-Feng Tan, and members of the Swartz laboratory for helpful discussions. This research was supported by the Intramural Research Program of the National Institute of Neurological Disorders and Stroke, NIH, Bethesda, MD to KJS (NS002945-28).

## Additional information

### Competing interests

Kenton J Swartz: Reviewing editor, *eLife*. The other authors declare that no competing interests exist.

### Funding

| Funder | Grant reference number | Author |
| --- | --- | --- |
| National Institute of Neurological Disorders and Stroke | NS002945-28 | Kenton J Swartz |

The funders had no role in study design, data collection, and interpretation, or the decision to submit the work for publication.

### Author contributions

Kate Huffer, Conceptualization, Data curation, Formal analysis, Validation, Investigation, Visualization, Methodology, Writing – original draft, Writing – review and editing; Matthew CS Denley, Data curation, Formal analysis, Investigation, Writing – review and editing; Elisabeth V Oskoui, Data curation, Investigation, Writing – review and editing; Kenton J Swartz, Data curation, Formal analysis, Supervision, Validation, Investigation, Visualization, Writing – original draft, Project administration, Writing – review and editing

### Author ORCIDs

Kate Huffer http://orcid.org/0000-0001-5003-3140
Matthew CS Denley http://orcid.org/0000-0003-2798-9448
Elisabeth V Oskoui https://orcid.org/0009-0002-5714-5991
Kenton J Swartz https://orcid.org/0000-0003-3419-0765

Reviewer #2 (Public review): https://doi.org/10.7554/eLife.99643.3.sa1
Author response https://doi.org/10.7554/eLife.99643.3.sa2

## Additional files

### Supplementary files
• MDAR checklist

## Data availability

All data needed to evaluate the conclusions in the article are present in the article. Population data for the electrophysiological experiments is provided in Source data files for each figure. Raw current traces for individual cells will be made available on request.

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
