## [Editor Report · eLife Assessment]

In this **valuable** study, Huffer et al posit that non-cold sensing members of the TRPM subfamily of ion channels (e.g., TRPM2, TRPM4, TRPM5) contain a binding pocket for icilin that overlaps with the one found in the cold-activated TRPM8 channel. After examining a body of TRP channel cryo-EM structures to identify the conserved site, this study presents **convincing** electrophysiological evidence supporting the presence of an icilin binding pocket within TRPM4. This study shows that icilin has modulatory effects on the TRPM4 channel and will be of direct interest to those working in the TRP-channel field, but it also has implications for studies of somatosensation, taste, as well as pharmacological targeting of the TRPM subfamily.

---

## [Referee Report · Reviewer #2 (Public review)]

Summary:

The authors set out to study whether the cooling agent binding site in TRPM8, which is located between the S1-S4 and the TRP domain, is conserved within the TRPM family of ion channels. They specifically chose the TRPM4 channel as the model system, which is directly activated by intracellular Ca2+. Using electrophysiology, the authors characterized and compared the Ca2+ sensitivity and the voltage-dependence of TRPM4 channels in the absence and presence of synthetic cooling agonist icilin. They also analyzed the mutational effects of residues (A867G and R901H; equivalent mutations in TRPM8 were shown involved in icilin sensitivity) on Ca2+ sensitivity and voltage-dependence of TRPM4 in the absence and presence of Ca2+. Based on the results as well as structure/sequence alignment, the authors concluded that icilin likely binds to the same pocket in TRPM4 and suggested that this cooling agonist binding pocket is conserved in TRPM channels.

Strengths:

The authors gave a very thorough introduction of the TRPM channels. They have nicely characterized the Ca2+ sensitivity and the voltage-dependence of TRPM4 channels and demonstrated icilin potentiates the Ca2+ sensitivity and diminishes the outward rectification of TRPM4. These results indicate icilin modulates TRPM4 activation by Ca2+.

The authors have incorporated additional data analysis and control experiments in the revised manuscript to strengthen their findings. They have well addressed the concerns raised by reviewers in the responses.

Weaknesses:

The study is conducted based on an assumption that TRPM4 activation is controlled by Ca2+ binding to a single site in the S1-S4 pocket in each subunit, and the second Ca2+ site in the cytoplasmic MHRs is simplified.

Despite the technical reasons presented by the authors in the rebuttal, the conclusion of this study can be strengthened if more cooling compounds- the most well-studied natural cooling agonist menthol, and/or other cooling agonists such as WS-12 and/or C3-are tested for their effects on TRPM4 and several other TRPM channels.

---

## [Author Response]

The following is the authors’ response to the original reviews.

**Public Reviews:**

**Reviewer #1 (Public Review):**
In this important study, Huffer et al posit that non-cold sensing members of the TRPM subfamily of ion channels (e.g., TRPM2, TRPM4, TRPM5) contain a binding pocket for icilin which overlaps with the one found in the cold-activated TRPM8 channel.The authors identify the residues involved in icilin binding by analyzing the existing TRPM8-icilin complex structures and then use their previously published approach of structure-based sequence comparison to compare the icilin binding residues in TRPM8 to other TRPM channels. This approach uncovered that the residues are conserved in a number of TRPM members: TRPM2, TRPM4, and TRPM5. The authors focus on TRPM4, with the rationale that it has the simplest activation properties (a single Ca2+-binding site). Electrophysiological studies show that icilin by itself does not activate TRPM4, but it strongly potentiates the Ca2+ activation of TRPM4, and introducing the A867G mutation (the mutation that renders avian TRPM8 sensitive to icilin) further increases the potentiating effects of the compound. Conversely, the mutation of a residue that likely directly interacts with icilin in the binding pocket, R901H, results in channels whose Ca2+ sensitivity is not potentiated by icilin.The data indicate that, just like in TRPV channels, the binding pockets and allosteric networks might be conserved in the TRPM subfamily.The data are convincing, and the authors employ good experimental controls.

We appreciate the supportive feedback of this reviewer.

**Reviewer #2 (Public Review):**
Summary:The authors set out to study whether the cooling agent binding site in TRPM8, which is located between the S1-S4 and the TRP domain, is conserved within the TRPM family of ion channels. They specifically chose the TRPM4 channel as the model system, which is directly activated by intracellular Ca2+. Using electrophysiology, the authors characterized and compared the Ca2+ sensitivity and the voltage dependence of TRPM4 channels in the absence and presence of synthetic cooling agonist icilin. They also analyzed the mutational effects of residues (A867G and R901H; equivalent mutations in TRPM8 were shown involved in icilin sensitivity) on Ca2+ sensitivity and voltage-dependence of TRPM4 in the absence and presence of Ca2+. Based on the results as well as structure/sequence alignment, the authors concluded that icilin likely binds to the same pocket in TRPM4 and suggested that this cooling agonist binding pocket is conserved in TRPM channels.Strengths:The authors gave a very thorough introduction to the TRPM channels. They have nicely characterized the Ca2+ sensitivity and the voltage-dependence of TRPM4 channels and demonstrated icilin potentiates the Ca2+ sensitivity and diminishes the outward rectification of TRPM4. These results indicate icilin modulates TRPM4 activation by Ca2+.

We appreciate the supportive feedback of this reviewer.

Weaknesses:The reviewer has a few concerns. First, icilin alone (at 25µM) and in the absence of Ca2+ does not activate the TRPM4 channel. Have the authors titrated a wide range of icilin concentrations (without Ca2+ present) for TRPM4 activation? It raises the question that whether icilin is indeed an agonist for TRPM4 channel. This has not been tested so it is unclear. One may argue that icilin needs Ca2+ as a co-factor for channel activation just like in TRPM8 channel. This leads to the second concern, which is a complication in the experimental design and data interpretation. TRPM4 itself requires Ca2+ for activation to begin with, thus it is hard to dissect whether the current observed here for TRPM4 is activated by Ca2+ or by icilin plus its cofactor Ca2+. This is the difference between TRPM8 and TRPM4, as TRPM8 itself is not activated by Ca2+, thus TRPM8 activation is through icilin and Ca2+ acts as a prerequisite for icilin activation.

We agree that the comparison between TRPM8 and TRPM4 is not perfect because TRPM4 requires Ca2+ for activation, but it is clear that the current activated by Ca2+ in the presence of icilin also involves icilin because it activates at lower Ca2+ concentrations and lower voltages. We have tested icilin at concentrations between 12.5 and 25 µM and at these concentrations icilin does not activate TRPM4 when applied alone, so we have no evidence that it is an agonist. Both of these concentrations are higher than those reported by Chuang et al. to be saturating for TRPM8 in the presence of Ca2+. We haven’t tested icilin at higher concentrations because we wanted to keep the final concentration of DMSO low enough to avoid any effects of the vehicle. We now emphasize this even more clearly in the revised manuscript.

The results presented in this study are only sufficient to show that icilin modulates the Ca2+-dependent activation of TRPM4 and icilin at best may act as an allosteric modulator for TRPM4 function. One cannot conclude from the current work that icilin is an agonist or even specifically a cooling agonist for TRPM4. Icilin is a cooling agonist for TRPM8, but it does not mean that if icilin modulates TRPM4 activity then it serves as a cooling agonist for TRPM4.

We agree with these comments, and we believe that the intent of our statements in the manuscript are completely in line with this perspective. We never refer to icilin as a cooling agent for TRPM4 but rather refer to the cooling agent binding pocket in TRPM8 and how that appears to be conserved and functions in TRPM4 to modulate opening of the channel. We have carefully gone through the manuscript to refer directly to icilin by name (rather than as a cooling agent) when referring to its actions on TRPM4 to make sure there is no confusion.

For the mutation data on A867G, Figure 4A-B, left panels, it looks like A867G has stronger Ca2+ sensitivity compared to the WT in the absence of icilin and the onset of current activation is faster than the WT, or this is simply due to the scale of the data figure are different between A867G and the WT. Overall the mutagenesis data are weak to support the conclusion that icilin binds to the S1-S4 pocket. The authors need to mutate more residues that are involved in direct interaction with icilin based on the available structural information, including but limited to residues equivalent to Y745 and H845 in human TRPM8.

The A867G mutant does seem to promote opening by Ca2+ in the absence of icilin, and we now comment on this in the manuscript. Having said that, we have not carefully studied the concentration-dependence for activation by Ca2+ because at higher concentrations we see evidence of desensitization. We think Ca2+, icilin and depolarized voltages promote an open state of TRPM4 and the A867G does so as well.

We respectfully disagree about the strength of mutagenesis results present in our manuscript. We present clear gain and loss of function for two mutants corresponding to influential residues within the cooling agent binding pocket of TRPM8. We agree that Y786 mutations would have been a valuable addition, and our plan was to include mutations of this residue. Unfortunately, both the Y786A and Y786H mutants exhibited rundown to repeated stimulation by Ca2+, making them challenging to obtain reliable results on their effects on modulation by icilin.

The authors set out to study the conservation of the cooling agonist binding site in TRPM family, but only tested a synthetic cooling agonist icilin on TRPM4. In order to draw a broad conclusion as the title and the discussion have claimed, the authors need to more cooling compounds, including the most well-known natural cooling agonist menthol, and other cooling agonists such as WS-12 and/or C3, and test their effects on several TRPM channels, not just TRPM4. With the current data, the authors need to significantly tone down the claim of a conserved cooling agonist binding pocket in the TRPM family.

We would have liked to broaden the scope to other ligands that modulate TRPM8 and we agree that including those data would certainly reinforce our conclusions. However, the first author recently moved on to a new faculty position and extending our findings would require enlisting another member of the lab and take away from their independent projects. We also do not agree that this is essential to support any of our conclusions. It is also important to keep in mind that icilin is a high-affinity ligand for TRPM8, such that weaker interactions with TRPM4 can still be readily observed. We think it is likely that lower affinity agonists like menthol might not have sufficient affinity to see activity in TRPM4. This scenario is not unlike our earlier experience with TRPV channels where we succeeded in engineering vanilloid sensitivity into TRPV2 and TRPV3 using the high affinity agonist resiniferatoxin (Zhang et al., 2016, eLife). In the case of TRPV2, another group had made the same quadruple mutant and failed to see activation by capsaicin even though resiniferatoxin also worked in their hands (see Fig. 2 in Yang et al., 2016, PNAS).

On page 11, the authors suggest based on the current data, that TRPM2 and TRPM5 may also be sensitive to cooling agonists because the key residues are conserved. TRPM2 is the closest homolog to TRPM8 but is menthol-insensitive. There are studies that attempted to convert menthol sensitivity to TRPM2, for example, Bandell 2006 attempted to introduce S2 and TRP domains from TRPM8 into TRPM2 but failed to make TRPM2 a menthol-sensitive channel. The sequence conservation or structural similarity is not sufficient for the authors to suggest a shared cooling agonist sensitivity or even a common binding site in the TRPM2 and TRPM5 channels. Again, as pointed out above, the authors need to establish the actual activation of other TRPM channels by these agonists first, before proceeding to functionally probe whether other TRPM channels adopt a conserved agonist binding site.

We are somewhat confused by these comments because we do not comment about whether cooling agents can activate TRPM2 or TRPM5. We simply analyzed the structures to make the point that the key residues in the cooling agent binding pocket of TRPM8 are conserved in these other TRPM channels. The Bandell paper is relevant, but it is also possible that they failed to uncover a relationship because they only used an agonist that has relatively low affinity for TRPM8. It would have been interesting to see what they might have found if they had used a high-affinity ligand like icilin instead of a low affinity ligand like menthol.

Taken together, this current work presents data to show the modulatory effects of icilin on the Ca2+ dependent activation and voltage dependence of the TRPM4 channel.

We agree.

**Reviewer #3 (Public Review):**
Summary:The family of transient receptor potential (TRP) channels are tetrameric cation selective channels that are modulated by a variety of stimuli, most notably temperature. In particular, the Transient receptor potential Melastatin subfamily member 8 (TRPM8) is activated by noxious cold and other cooling agents such as menthol and icilin and participates in cold somatosensation in humans. The abundance of TRP channel structural data that has been published in the past decade demonstrates clear architectural conservation within the ion channel family. This suggests the potential for unifying mechanisms of gating despite their varied modes of regulation, which are not yet understood. To address this question, the authors examine the 264 structures of TRP channels determined to date and observe a potential binding pocket for icilin in multiple members of the Melastatin subfamily, TRPM2, TRPM4, and TRPM5. Interestingly, none of the other Melastatin subfamily members had been shown to be sensitive to icilin apart from TRPM8. Each of these channels is activated by intracellular calcium (Ca2+) and a Ca2+ binding site neighbors the predicted pocket for icilin binding in all cryo-EM structures. The authors examined whether icilin could modulate the activation of TRPM4 in the presence of intracellular Ca2+. The addition of icilin enhances Ca2+-dependent activation of TRPM4, promotes channel opening at negative membrane potentials, and improves the kinetics of opening. Furthermore, mutagenesis of TRPM4 residues within the putative icilin binding pocket predicted to enhance or diminish TRPM4 activity elicit these behaviors. Overall, this study furthers our understanding of the Melastatin subfamily of TRP channel gating and demonstrates that a conserved binding pocket observed between TRPM4 and TRPM8 channel structures can function similarly to regulate channel gating.Strengths:This is a simple and elegant study capitalizing on a vast amount of high-resolution structural information from the TRP channel of ion channels to identify a conserved binding pocket that was previously unknown in the Melastatin subfamily, which is interrogated by the authors through careful electrophysiology and mutagenesis studies.Weaknesses:No weaknesses were identified by this reviewer.

We appreciate the supportive comments of the review.

**Recommendations for the authors:**

**Reviewer #1 (Recommendations For The Authors):**
I don't have any major asks, but a few questions did arise while reading your work.(1) You refer multiple times to the VSLD pocket as being "open to the cytoplasm". It is not clear if you are implying that compounds such as icilin access the pocket via the cytoplasm (e.g., permeate the membrane to the cytosol, and then enter the binding site?) Is there data to support this? Some clarification here would be helpful, and perhaps explain if there is any distinction between how calcium might enter the VSLD binding site vs hydrophobic compounds like icilin.

This is an excellent point. Our reference to “open to the cytoplasm” was for Ca2+ ions and we have no evidence for how icilin enters the cooling agent binding pocket. We had tried to look for evidence that Ca2+ might trap icilin in TRPM4 but at the end of the day the results were not convincing enough to include in the manuscript. We have added data showing that icilin slows deactivation of TRPM4 after removing Ca2+, which is particularly evident in the A867G mutant, but this doesn’t inform on whether Ca2+ can trap icilin. We have added a statement about not knowing how icilin enters or leaves the cooling agent binding pocket in TRPM channels.

(2) Icilin is referred to as a "cooling compound", but its cooling effects are dependent on its interactions with TRPM8. This might be something to clarify, as it might otherwise be understood that other TRPM channels that interact with icilin also mediate the sensing of cool temperatures.

This is another excellent point and we have no reason to believe that icilin interacting with any TRPM channel other than TRPM8 mediates cooling sensations. We have added a statement to this effect in the discussion when considering actions of icilin that might be mediated by TRPM4 channels.

**Reviewer #2 (Recommendations For The Authors):**
(1) The title and statements in the results/discussion refer to icilin as a cooling agonist of TRPM4 and binds to a conserved "cooling agonist binding pocket", and the authors suggested a similar role and binding site for icilin in TRPM2 and TRPM5 channel. It is a too broad conclusion that is not fully supported by the current experimental data, which only shows icilin works as a modulator, not an agonist for TRPM4 channel. The authors should change the usage of cooling agonist or conserved cooling agonist binding pocket plus significantly tone down the conclusion of a conserved cooling agonist binding pocket, which is potentially misleading. Alternatively, if the authors insist on using cooling agonist in this context, they should establish the activation of TRPM4, TRPM2, and TRPM5 by icilin as the first step, because the current data only support icilin as a TRPM4 modulator but not an agonist.

We respectfully don’t agree with this opinion. We show broad conservation of the cooling agent binding pocket in structures of many TRPM channels, and we chose one of them to test for a functional relationship. We think that the title accurately reflects the topic of the paper and does not specify the extent to which functional conservation has been demonstrated and we would like to keep it. The distinction between agonist and modulator is not even germane because icilin is not an agonist of TRPM8 either.

(2) The manuscript will be strengthened if the authors test additional cooling compounds of TRPM8, including menthol, the menthol analog WS-12, and C3. More importantly, distinct from icilin, these three compounds do not depend on Ca2+ to activate the TRPM8 channel. Thus when testing these compounds on TRPM4, it may reduce the complication of the role of Ca2+, as TRPM4 channel itself requires Ca2+ for activation.

We restate our response to this point in the public review…

We would have liked to broaden the scope to other ligands that modulate TRPM8 and we agree that including those data would certainly reinforce our conclusions. However, the first author recently moved on to a new faculty position and extending our findings would require enlisting another member of the lab and taking away from their independent projects. We also do not agree that this is essential to support any of our conclusions. It is also important to keep in mind that icilin is a high-affinity ligand for TRPM8, such that weaker interactions with TRPM4 can still be readily observed. We think it is likely that lower affinity agonists like menthol might not have sufficient affinity to see activity in TRPM4 This scenario is not unlike our earlier experience with TRPV channels where we succeeded in engineering vanilloid sensitivity into TRPV2 and TRPV3 using the high affinity agonist resiniferatoxin (Zhang et al., 2016, eLife). In the case of TRPV2, another group had made the same quadruple mutant and failed to see activation by capsaicin even though resiniferatoxin also worked in their hands (see Fig. 2 in Yang et al., 2016, PNAS).

(3) The manuscript will be strengthened if the authors test additional residues in the S1-S4 pocket that form direct interactions or are within interacting distances with icilin based on the cryo-EM structures.

We restate our response to this point in the public review…

We present clear gain and loss of function for two mutants corresponding to influential residues within the cooling agent binding pocket of TRPM8. We agree that Y786 mutations would have been a valuable addition and our plan was to include mutations of this residue. Unfortunately, both the Y786A and Y786H mutants exhibited rundown, making them challenging to obtain reliable results on their effects on modulation by icilin.

Furthermore, the ambiguity in the icilin binding pose based on available TRPM8 structures complicates structure-based identification of the most important interacting residues in TRPM8, and we would have needed to functionally validate the effects of any novel mutations we identified in TRPM8 prior to testing them in TRPM4. Instead, we have based our mutagenesis on constructs that have been previously characterized to affect the sensitivity of TRPM8 to cooling agents. A systematic mutagenesis scan of TRPM8 residues predicted to interact differentially with icilin in the two different available binding poses would likely help clarify the true binding pose of icilin and would be an interesting future study.

**Reviewer #3 (Recommendations For The Authors):**
I enjoyed reading this manuscript. It was well-executed and written. It will be interesting to corroborate these findings with a cryo-EM structure of TRPM2, TRPM4, or TRPM5 in the presence of icilin.

We agree and may pursue these in future studies. This would be particularly interesting given ambiguities in how icilin docks into TRPM8 in previously published structures.

Minor comments/questions:Have the authors considered icilin accessibility to its binding pocket? In other words, could the presence of intracellular Ca2+ inhibit the accessibility of icilin to its binding pocket in TRPM4? It should be a straightforward experiment, I think it would be informative, and could further support the authors' conclusion of the location of the TRPM4 icilin binding pocket.

We completely agree and we had tried to look for evidence that Ca2+ might trap icilin in TRPM4 but at the end of the day the results were not convincing enough to include in the manuscript. We have added data showing that icilin slows deactivation of TRPM4 after removing Ca2+, which is particularly evident in the A867G mutant, but this doesn’t inform on whether Ca2+ can trap icilin. We have added a statement about not knowing how icilin enters or leaves the cooling agent binding pocket in TRPM channels.

Figures 7 and 8 are missing the 0 µM Ca2+ control trace in the presence of 25 µM icilin.

All sample traces from Figures 7 and 8 are shown from a single cell for the sake of comparison (Likewise, the sample traces from Figures 3 and 4 come from a single cell, and the sample traces from Figures 5 and 6 come from a single cell). Unfortunately, we were unable to obtain data from an R901H mutant cell that contained all six conditions we wished to show, and there is no representative trace for 0 µM Ca2+ in the presence of 25 µM icilin for that cell.

This is up to the discretion of the authors, but perhaps a better way to arrange the paper Figures would be to combine Figures 5-6 and Figures 7-8 and rearrange the data to place some in a supplementary figure (e.g. Figure 5-6 = Figure 5 and Figure 5 - Figure Supplement 1, Figure 7-8 = Figure 6 and Figure 6 - Figure Supplement 1).

We carefully considered these suggestions and we appreciate the reviewers’ flexibility but would prefer to retain the original arrangement of data in the figures.

Are there any mutations in the icilin binding pocket in TRPM4, and presumably TRPM2 and TRPM5, that are associated with human disease? This is a question that came to my mind and not one that needs to be addressed in the manuscript.

This is an interesting point. There are quite a few disease-associated mutants within TRPM4 at positions corresponding to the cooling agent binding pocket in TRPM8. We could not see an appropriate place in the discussion where we could concisely bring this information in so we decided against commenting.